# Scalable Thompson Sampling using Sparse Gaussian Process Models

**Sattar Vakili**[*1], **Henry Moss**[*2], **Artem Artemev**[2,3], **Vincent Dutordoir**[2,4], **Victor Picheny**[2]

## Abstract

Thompson Sampling (TS) from Gaussian Process (GP) models is a powerful tool for the optimization of black-box functions. Although TS enjoys strong theoretical guarantees and convincing empirical performance, it incurs a large computational overhead that scales polynomially with the optimization budget. Recently, scalable TS methods based on sparse GP models have been proposed to increase the scope of TS, enabling its application to problems that are sufficiently multi-modal, noisy or combinatorial to require more than a few hundred evaluations to be solved. However, the approximation error introduced by sparse GPs invalidates all existing regret bounds. In this work, we perform a theoretical and empirical analysis of scalable TS. We provide theoretical guarantees and show that the drastic reduction in computational complexity of scalable TS can be enjoyed without loss in the regret performance over the standard TS. These conceptual claims are validated for practical implementations of scalable TS on synthetic benchmarks and as part of a real-world high-throughput molecular design task.

## 1 Introduction

Thompson sampling [TS, 1] is a popular algorithm for Bayesian optimization [BO, 2] — a sequential model-based approach for the optimization of expensive-to-evaluate black-box functions, typically characterised by limited prior knowledge and access to only a limited number of (possibly noisy) evaluations. By sequentially evaluating the maxima of random samples from a model of the objective function, TS provides a conceptually simple method for balancing exploration and exploitation.

TS is often paired with Gaussian Processes (GPs), which offers a spectrum of powerful and flexible modeling tools that provide probabilistic predictions of the objective function. The resulting GP-TS algorithms [3] have been found to provide highly efficient optimization under heavily restricted optimization budgets, with numerous successful applications including aerodynamic design [4], route planning [5] and web-streaming [6]. While most popular BO algorithms cannot query more than a handful of points at a time [7–10] without employing replicating designs [see 11, 12], TS has a natural ability to query large batches of points. Therefore, TS is a popular solution for optimization pipelines enjoying a large degree of parallelisation, for example in high-throughout chemical space exploration [13] and for the distributed tuning of machine learning models across cloud compute resources [14].

As BO incurs a substantial computational overhead between successive iterations, while updating models and choosing the next set of query points, standard BO methods are limited to optimization problems with small evaluation budgets [2]. However, with large batches, the computational overhead incurred by BO per individual function evaluation is considerably reduced. Therefore, considering large batches is a promising tactic to expand BO to larger optimization budgets, which are required to

---

[*]Equal contribution, [1] MediaTek Research, [2] Secondmind, [3] Imperial College London, [4] University of Cambridge. Correspondence to Sattar Vakili <sattar.vakili@mtkresearch.com>, Henry Moss <henry.moss@secondmind.ai>.

optimize highly noisy problems with rougher optimization landscapes [11, 12] or high dimensional and combinatorial search spaces [15, 13, 16]. Consequently, the highly-parallelizable TS is a promising candidate for BO under large optimization budgets.

Unfortunately, practical implementations of GP-TS suffer from two key computational bottlenecks that prevent the method from scaling in terms of total optimization budget. Not only does each update of the GP posterior distribution require a matrix inversion that incurs a cubic cost w.r.t. the number of observations $t$ [17], but even sampling from this posterior can be a daunting task — the standard approach of drawing a joint sample across a $N$ point discretization of the search space has an $O(N^3)$ complexity [due to a Cholesky decomposition step, 18]. Alternative existing approaches for BO under large optimization budgets include using Neural Networks in lieu of GPs [15, 13] or to use local models [19] and ensembles [16].

A natural answer to the scalability issues of GP-TS is to rely on the recent advances in Sparse Variational GP models [SVGP, 20]. SVGPs provide a low rank $O(m^2 t)$ approximation of the GP posterior, where $m$ is the number of the so-called *inducing variables* that grows at a rate much slower than $t$. Successful applications of SVGPs for BO under large optimization budgets include optimizing a free-electron laser [21], molecules under synthesis-ability constraints [22], and the composition of alloys [23]. Furthermore, [24] introduced an efficient sampling rule (referred to as *decoupled sampling*) which can be used to efficiently perform TS with SVGPs. In particular, [24] decomposes samples from the SVGP posterior into the sum of an approximate prior based on $M$ features (see Sec. 3.3) and an SVGP model update, thus reducing the computational cost of drawing a Thompson sample to $O((m + M)N)$. Leveraging this sampling rule results in a scalable GP-TS algorithm (henceforth S-GP-TS) that can handle orders of magnitude greater optimization budgets.

While [3] proposed a comprehensive theoretical analysis of exact GP-TS, it does not apply to S-GP-TS. Indeed, using sparse models and decoupled sampling introduce two layers of approximation, that must be handled with care, as even a small constant error in the posterior can lead to poor performance by encouraging under-exploration in the vicinity of the optimum point [25]. Our primary contributions can be summarised as follows. First, we provide a theoretical analysis showing that batch TS from any approximate GP can achieve the same regret order as an exact GP-TS algorithm as long the quality of the posterior approximations satisfies certain conditions (Assumptions 3 and 4). Second, for the specific case of S-GP-TS (batch decoupled TS using a SVGP), we leverage the results of [26] to provide bounds in terms of GP's kernel spectrum for the number of prior features and inducing variables required to guarantee low regret. Finally, we investigate empirically the performance of multiple practical implementations of S-GP-TS, considering synthetic benchmarks and a high-throughput molecular design task.

## 2 Problem Formulation

We consider the sequential optimization of an unknown function $f$ over a compact set $\mathcal{X} \subset \mathbb{R}^d$. A sequential learning policy selects a batch of $B$ observation points $\{x_{t,b}\}_{b \in [B]}$ at each time step $t = 1, 2, \ldots, T$ and receives the corresponding real-valued and noisy rewards $\{y_{t,b} = f(x_{t,b}) + \epsilon_{t,b}\}_{b \in [B]}$, where $\epsilon_{t,b}$ denotes the observation noise. Throughout the paper, we use the notation $[n] = \{1, 2, \ldots, n\}$, for $n \in \mathbb{N}$. As is common in both the bandits and GP literature, our analysis uses the following sub-Gaussianity assumption, a direct consequence of which is that $\mathbb{E}[\epsilon_{t,b}] = 0$, for all $t, b \in \mathbb{N}$.

**Assumption 1.** *$\epsilon_{t,b}$ are i.i.d., over both $t$ and $b$, $R-$sub-Gaussian random variables, where $R > 0$ is a fixed constant. Specifically, $\mathbb{E}[e^{h\epsilon_{t,b}}] \leq \exp(\frac{h^2 R^2}{2})$, $\forall h \in \mathbb{R}, \forall t, b \in \mathbb{N}$.*

Let $x^* \in \operatorname{argmax}_{x \in \mathcal{X}} f(x)$ be an optimal point. We can then measure the performance of a sequential optimizer by its *strict regret*, defined as the cumulative loss compared to $f(x^*)$ over a time horizon $T$

$$R(T, B; f) = \mathbb{E}\left[\sum_{t=1}^{T} \sum_{b=1}^{B} f(x^*) - f(x_{t,b})\right], \tag{1}$$

where the expectation is with respect to the randomness in noise and the possible stochasticity in the sequence of the selected batch observation points $\{x_{t,b}\}_{t \in [T], b \in [B]}$. Note that our regret measure (1) is defined for the true unknown $f$. In contrast, the alternative Bayesian regret [see e.g. 27, 14] averages over a prior distribution for $f$. As upper bounds on strict regret directly apply to the Bayesian

regret (but not necessarily the reverse), our results are stronger than those that can be achieved when analysing just Bayesian regret, for example when applying the technique of [28] that equates TS's Bayesian regret with that of the well-studied upper confidence bound policies.

Following [3, 29, 30], our analysis assumes a regularity condition on the objective function motivated by kernelized learning models and their associated reproducing kernel Hilbert spaces [RKHS, 31]:

**Assumption 2.** *Given an RKHS $H_k$, the norm of the objective function is bounded: $||f||_{H_k} \leq \mathcal{B}$, for some $\mathcal{B} > 0$, and $k(x, x') \leq 1$, for all $x, x' \in \mathcal{X}$.*

In the case of practically relevant kernels, Assumption 2 implies certain smoothness properties for the objective functions.

# 3 Gaussian Processes and Sparse Models

GPs are powerful non-parametric Bayesian models over the space of functions [17] with a distribution specified by a mean function $\mu(x)$ (henceforth assumed to be zero for simplicity) and a positive definite kernel (or covariance function) $k(x, x')$. We provide here a brief description of the classical GP model and two sparse variational formulations.

## 3.1 Exact Gaussian Process models

Suppose that we have collected a set of location-observation tuples $\mathcal{H}_t = \{\mathbf{X}_t, \mathbf{y}_t\}$, where $\mathbf{X}_t$ is the $tB \times d$ matrix of locations with rows $[\mathbf{X}_t]_{(s-1)B+b} = x_{s,b}$, and $\mathbf{y}_t$ is the $tB$-dimensional column vector of observations with elements $[\mathbf{y}_t]_{(s-1)B+b} = y_{s,b}$, for all $s \in [t], b \in [B]$. Then, assuming a Gaussian observation noise , the posterior of the GP model $\hat{f}$ given the set of past observations $\mathcal{H}_t$, is also a GP with mean $\mu_t(\cdot)$, variance $\sigma_t^2()$ and kernel function $k_t(\cdot, \cdot)$ specified as

$$\mu_t(x) = k_{\mathbf{X}_t,x}^{\mathbf{T}}(K_{\mathbf{X}_t,\mathbf{X}_t} + \tau\mathbf{I})^{-1}\mathbf{y}_t, \quad k_t(x, x') = k(x, x') - k_{\mathbf{X}_t,x}^{\mathbf{T}}(K_{\mathbf{X}_t,\mathbf{X}_t} + \tau\mathbf{I})^{-1}k_{\mathbf{X}_t,x'}, \quad (2)$$

and $\sigma_t^2(x) = k_t(x, x)$, with $k_{\mathbf{X}_t,x}$ the $tB$ dimensional column vector with entries $[k_{\mathbf{X}_t,x}]_{(s-1)B+b} = k(x_{s,b}, x)$, and $K_{\mathbf{X}_t,\mathbf{X}_t}$ the $tB \times tB$ positive definite covariance matrix with entries $[K_{\mathbf{X}_t,\mathbf{X}_t}]_{(s-1)B+b,(s'-1)B+b'} = k(x_{s,b}, x_{s',b'})$. We directly see from (2) that accessing the posterior expressions require an $O((tB)^3)$ matrix inversion, which is a computational bottleneck for large values of $tB$.

Note that in our problem formulation $f$ is fixed and observation noise has an unknown sub-Gaussian distribution. Using a GP prior and assuming a Gaussian noise is merely for ease of modelling and does not affect our assumptions on $f$ and $\epsilon_{t,b}$. The notation $\hat{f}$ is thus used to distinguish the GP model from the fixed $f$.

## 3.2 Sparse Variational Gaussian Process Models with Inducing Points

To overcome the cubic cost of exact GPs, SVGPs [20, 32] instead approximate the GP posterior through a set of *inducing points* $\mathbf{Z}_t = \{z_1, ..., z_{m_t}\}$ ($z_i \in \mathcal{X}$, with $m_t << tB$). Conditioning on the *inducing variables* $\mathbf{u}_t = \hat{f}(\mathbf{Z}_t)$ (rather than the $tB$ observations in $\mathbf{y}_t$) and specifying a prior Gaussian density $q_t(\mathbf{u}_t) = \mathcal{N}(\mathbf{m}_t, \mathbf{S}_t)$, yields an approximate posterior distribution that, crucially, is still a GP but with the significantly reduced computational complexity of $O(m_t^2 t)$. The posterior mean and covariance of the SVGP are given in closed form as

$$\mu_t^{(s)}(x) = k_{\mathbf{Z}_t,x}^{\mathbf{T}} K_{\mathbf{Z}_t,\mathbf{Z}_t}^{-1} \mathbf{m}_t \quad k_t^{(s)}(x, x') = k(x, x') + k_{\mathbf{Z}_t,x}^{\mathbf{T}} K_{\mathbf{Z}_t,\mathbf{Z}_t}^{-1} (\mathbf{S}_t - K_{\mathbf{Z}_t,\mathbf{Z}_t}) K_{\mathbf{Z}_t,\mathbf{Z}_t}^{-1} k_{\mathbf{Z}_t,x'}.$$

The variational parameters $\mathbf{m}_t$ and $\mathbf{S}_t$ are set as the maximizers of the evidence lower bound (ELBO, see Appendix A for the details) and can be optimized numerically with mini-batching [32]. There are various standard ways in practice to select the locations of the inducing points $\mathbf{Z}_t$, e.g. by using an experimental design, sampling from a k-DPP (that stands for determinantal point process), or by optimizing them along with the inducing variables.

## 3.3 Sparse Variational Gaussian Process Models with Inducing Features

An alternative approximation strategy is using inducing feature approximations [33, 26, 34]. Here, we define inducing variables as the linear integral transform of $\hat{f}$ with respect to some *inducing*

*features* [35] $\psi_1(x), .., \psi_{m_t}(x)$, i.e we set our $i^{\text{th}}$ inducing variable as $u_{t,i} = \int_{\mathcal{X}} \hat{f}(x)\psi_i(x)dx$. Courtesy of Mercer's theorem, we can usually decompose our chosen kernel $k$ as the inner product of possibly infinite dimensional feature maps (see Theorem 4.1 in [36]) to provide the expansion $k(x, x') = \sum_{j=1}^{\infty} \lambda_j \phi_j(x).\phi_j(x')$ for eigenvalues $\{\lambda_j \in \mathbb{R}^+\}_{j=1}^{\infty}$ and eigenfunctions $\{\phi_j \in H_k\}_{j=1}^{\infty}$. If we set our inducing features to be the $m_t$ eigenfunctions with largest eigenvalues, it can be shown that $\text{cov}(u_{t,i}, u_{t,j}) = \lambda_j \delta_{i,j}$ and $\text{cov}(u_{t,j}, \hat{f}(x)) = \lambda_j \phi_j(x)$ , yielding an approximate Gaussian Process model with posterior mean and covariance given by

$$\mu_t^{(s)}(x) = \boldsymbol{\phi}_{m_t}^{\text{T}}(x)\mathbf{m}_t \qquad k_t^{(s)}(x, x') = k(x, x') + \boldsymbol{\phi}_{m_t}^{\text{T}}(x)(\mathbf{S}_t - \Lambda_{m_t})\boldsymbol{\phi}_{m_t}(x').$$

Here, $\mathbf{m}_t$ and $\mathbf{S}_t$ are inducing parameters (as above), $\boldsymbol{\phi}_{m_t}(x) \triangleq [\phi_1(x), ..., \phi_{m_t}(x)]^{\text{T}}$ is the truncated feature vector and $\Lambda_{m_t}$ is the $m_t \times m_t$ diagonal matrix of eigenvalues, $[\Lambda_{m_t}]_{i,j} = \lambda_i \delta_{i,j}$.

Inducing feature approximations have strong advantages, in particular a reduced computational cost and the fact that no inducing points need to be specified. However, accessing these eigenfeatures require the Mercer decomposition of the used kernel, which is available for certain kernels on manifolds [37, 34], but limited to low dimensions for others [38, 39].

## 4 Scalable Thompson Sampling using Gaussian Process Models (S-GP-TS)

At each BO step $t$, GP-TS proceeds by drawing $B$ i.i.d. samples $\{\hat{f}_{t,b}\}_{b \in [B]}$ from the posterior distribution of $\hat{f}$ and finding their maximizers, i.e. we select samples $x_{t,b}$ satisfying

$$\{x_{t,b} = \text{argmax}_{x \in \mathcal{X}} \hat{f}_{t,b}(x)\}_{b \in [B]}. \tag{3}$$

However, since $\hat{f}_{t,b}$ is an infinite dimensional object, generating such samples is computationally challenging. Consequently, it is common to resort to approximate strategies, the most simple of which is to sample across an $N_t$ point discretization $D_t$ of $\mathcal{X}$ [14] which can be obtained with an $O(N_t^3)$ cost (due to a required Cholesky decomposition).

To improve the computational efficiency of TS, a classical strategy [40, 41] is to rely on kernel decompositions. For instance, a sample $\hat{f}$ from a GP can be expressed as a randomly weighted sum of the kernel's eigenfunctions $\hat{f}(x) = \sum_{j=1}^{\infty} \sqrt{\lambda_j} w_j \phi_j(x)$, or, in the case of shift-invariant kernels, the kernel's Fourier features $\psi_j(x)$ (see [42]) as $\hat{f}(x) = \sum_{j=1}^{\infty} w_j \psi_j(x)$. By truncating these infinite expansions to contain only the $M$ eigenfunctions with largest eigenvalues or $M$ random Fourier features, we have access to approximate but analytically tractable samples. For both expansions, the weights $w_j$ are sampled independently from a standard normal distribution. Conditioned on current $tB$ observations, the posterior distribution of $w_j$ are Gaussian with mean and covariance functions that can be calculated with an $O(M^3)$ computations, resulting in an $O(M^3 + BNM)$ cost to draw $B$ Thompson samples.

Fast approximation strategies described above avoid costly matrix operations and work best only when sampling from GP priors. Posterior GP distributions are often too complex to be well-approximated by a finite feature representation [16, 43, 30]. The recent work of [24] tackled this issue by using truncated feature representations only to approximate the prior GP and a separate model update term to approximate posterior samples. For SVGP models, this has been shown to yield more accurate Thompson samples whilst incurring only an $O((m_t + M)BN)$, on top of the $O(tBm_t^2)$ SVGP model fit, per optimization step $t$.

For our theoretical analysis, we consider two distinct decoupled sampling rules inspired by [24] , one for each of the two SVGP formulations presented above [see 24,  for derivations and similar expressions for Fourier decompositions]. The first rule is referred to as *Decoupled Sampling with Inducing Points* and is defined as

$$\tilde{f}_t(x) = \sum_{j=1}^{M} \alpha_t \sqrt{\lambda_j} w_j \phi_j(x) + \sum_{j=1}^{m_t} v_{t,j} k(x, z_j), \tag{4}$$

where we have coefficients $v_{t,j} = [K_{\mathbf{Z}_t, \mathbf{Z}_t}^{-1}(\alpha_t(\mathbf{u}_t - \mathbf{m}_t) + \mathbf{m}_t - \alpha_t \boldsymbol{\Phi}_{m_t, M} \Lambda_M^{\frac{1}{2}} \mathbf{w}_M)]_j$ for $\boldsymbol{\Phi}_{m_t, M} = [\boldsymbol{\phi}_M(z_1), ..., \boldsymbol{\phi}_M(z_{m_t})]^{\text{T}}$ and $\mathbf{w}_M = [w_1, ..., w_M]^{\text{T}}$. The weights $w_i$ are drawn i.i.d from $\mathcal{N}(0, 1)$.

([4](#)) is a modification of the sampling rule of [24] where we have added a scaling parameter $\alpha_t \in \mathbb{R}$ (with $\alpha_t = 1$, the sampling rule of [24] is recovered). When set to be greater than one, $\alpha_t$ serves to increases the variability of the approximate function samples (without changing their mean) and is used in our analysis to ensure sufficient exploration.

To efficiently sample from our second class of SVGP models, we also consider *Decoupled Sampling with Inducing Features*:

$$\tilde{f}_t(x) = \sum_{j=1}^{M} \alpha_t \sqrt{\lambda_j} w_j \phi_j(x) + \sum_{j=1}^{m_t} v_{t,j} \lambda_j \phi_j(x), \tag{5}$$

where $v_{t,j} = [\Lambda_{m_t}^{-1}(\alpha_t(\mathbf{u}_t - \mathbf{m}_t) + \mathbf{m}_t - \alpha_t \Lambda_{m_t}^{\frac{1}{2}} \mathbf{w}_{m_t})]_j$ for $\Lambda_{m_t}$ defined in Section [3.3](#).

## 5 Regret Analysis of S-GP-TS

Here, we first establish an upper bound on the regret of any approximate GP model (Theorem [1](#)) based on the quality of their approximate posterior, as parameterized in Assumptions [3](#) and [4](#). We then discuss the consequences of Theorem [1](#) for the regret bounds and the computational complexity of S-GP-TS methods based on SVGPs and the decoupled sampling rules ([4](#)) and ([5](#)).

### 5.1 Regret Bounds Based on the Quality of Approximations

Consider a TS algorithm using an approximate GP model. In particular, assume an approximate model is provided where $\tilde{k}_t$, $\tilde{\sigma}_t$ and $\tilde{\mu}_t$ are approximations of $k_t$, $\sigma_t$ and $\mu_t$, respectively. At each time $t$, a batch of $B$ samples $\{\tilde{f}_{t,b}\}_{b=1}^{B}$ is drawn from a GP with mean $\tilde{\mu}_{t-1}$ and the scaled covariance $\alpha_t^2 \tilde{k}_{t-1}$. The batch of observation points $\{x_{t,b}\}_{b=1}^{B}$ are selected as the maximizers of $\{\tilde{f}_{t,b}\}_{b=1}^{B}$ over a discretization $D_t$ of the search space.

We start our analysis by making two assumptions on the *quality* of approximations $\tilde{\mu}_t$, $\tilde{\sigma}_t$ of the posterior mean and the standard deviation. This parameterization is agnostic to the particular sampling rule (governing $\tilde{\mu}_t$ and $\tilde{\sigma}_t$) and provides valuable intuition that can be applied to any approximate method. When it comes to S-GP-TS (as the model governing $\tilde{\mu}_t$, $\tilde{\sigma}_t$), we show, in Sec. [5.2](#), that these assumptions are satisfied under some conditions on the value of the parameters of the sampling rules.

**Assumption 3** (quality of the approximate standard deviation)**.** *For the approximate $\tilde{\sigma}_t$, the exact $\sigma_t$, and for all $x \in \mathcal{X}$,*

$$\frac{1}{\underline{a}_t} \sigma_t(x) - \epsilon_t \leq \tilde{\sigma}_t(x) \leq \bar{a}_t \sigma_t(x) + \epsilon_t,$$

*where $1 \leq \underline{a}_t \leq \underline{a}$, $1 \leq \bar{a}_t \leq \bar{a}$ for all $t \geq 1$ and some constants $\underline{a}, \bar{a} \in \mathbb{R}$, and $0 \leq \epsilon_t \leq \epsilon$ for all $t \geq 1$ and some small constant $\epsilon \in \mathbb{R}$.*

**Assumption 4** (quality of the approximate prediction)**.** *For the approximate $\tilde{\mu}_t$, the exact $\mu_t$ and $\sigma_t$, and for all $x \in \mathcal{X}$,*

$$|\tilde{\mu}_t(x) - \mu_t(x)| \leq c_t \sigma_t(x),$$

*where $0 \leq c_t \leq c$ for all $t \geq 1$ and some constant $c \in \mathbb{R}$.*

The following Lemma establishes a concentration inequality for the approximate statistics using the one for exact statistics [3, Theorem 2].

**Lemma 1.** *Under Assumptions [1](#), [2](#), [3](#) and [4](#), with probability at least $1 - \delta$, $|f(x) - \tilde{\mu}_t(x)| \leq \tilde{u}_t(\tilde{\sigma}_t(x) + \epsilon_t)$, where $\tilde{u}_t(\delta) = \underline{a}_t \left( \mathcal{B} + R\sqrt{2(\gamma_{tB} + 1 + \log(1/\delta))} + c_t \right)$.*

Proof is provided in Appendix [B](#). Here, $\gamma_s$ is the *maximal information gain*: $\gamma_s = \max_{A \subset \mathcal{X}, |A|=s} \mathcal{I}([y(x)]_{x \in A}; [\hat{f}(x)]_{x \in A})$, where $\mathcal{I}([y(x)]_{x \in A}; [\hat{f}(x)]_{x \in A})$ denotes the mutual information [44, Chapter 2] between observations and the underlying GP model. The maximal information gain can itself be bounded for a specific kernel (see Sec. [5.3](#)).

Following [29] and [3], we consider a discretization $D_t$ of the search space satisfying the following assumption.

**Assumption 5.** *The discretization $D_t$ is designed in a way that $|f(x) - f(\mathbf{x}^{(t)})| \leq 1/t^2$ for all $x \in \mathcal{X}$, where $\mathbf{x}^{(t)} = argmin_{x' \in D_t}||x - x'||$ is the closest point (in Euclidean norm) to $x$ in $D_t$. The size of this discretization satisfies $|D_t| = N_t \leq C(d, B)t^{2d}$ where $C(d, B)$ is independent of $t$ ([3, 29]).*

We are now in a position to present regret bounds based on the quality of GP approximations:

**Theorem 1.** *Consider S-GP-TS with $\alpha_t = 2\tilde{u}_t(1/(t^2))$. Under Assumptions 1, 2, 3 , 4 and 5, the regret defined in (1), satisfies*

$$
\begin{aligned}
R(T, B; f) \quad &\leq \quad 30\bar{a}\beta_T B\sqrt{\frac{2T\gamma_T}{\log(1 + \frac{1}{\tau})}} + (31\beta_T + \alpha_T)\epsilon TB + 15B\mathcal{B} + 2B \\
&= O\left(\underline{a}\bar{a}BR\sqrt{d\gamma_T(\gamma_{TB} + \log(T))T\log(T)} + \underline{a}\epsilon TBR\sqrt{d(\gamma_{TB} + \log(T))\log(T)}\right), (6)
\end{aligned}
$$

*where $\beta_t = \alpha_t(b_t + \frac{1}{2})$ with $b_t = \sqrt{2\log(N_t t^2)}$.*

See the proof in Appendix B. This regret bound scales with the product of the ratios $\underline{a}$ and $\bar{a}$, with an additive term depending on the additive approximation error in the standard deviation.

## 5.2    Approximation Quality of the Decoupled Sampling Rule

For S-GP-TS with inducing points, we assume, as in [26], that the inducing points are sampled according to a discrete k-DPP. While this might be costly in practice, [26] showed that $\mathbf{Z}_t$ can be efficiently sampled from $\epsilon_0$ *close* sampling methods without compromising the predictive quality of SVGP. For both sampling rules, we also assume in our analysis that the Mercer decomposition of the kernel is used.

The quality of the approximation can be characterized using the spectral properties of the GP kernel. Let us define the tail mass of eigenvalues $\delta_M = \sum_{i=M+1}^{\infty} \lambda_i \bar{\phi}_i^2$ where $\bar{\phi}_i = \max_{x \in \mathcal{X}} \phi_i(x)$. With decaying eigenvalues, including sufficient eigenfunctions in the feature representation results in a small $\delta_M$. In addition, [26] showed that, for an SVGP, a sufficient number of inducing variables ensures that the Kullback–Leibler (KL) divergence between the approximate and the true posterior distributions diminishes. Consequently, the approximate posterior mean and the approximate posterior variance converge to the true ones. Building on this result, we are able to prove Proposition 1 on the quality of approximations.

**Proposition 1.** *For S-GP-TS based on sampling rule (4) with $\alpha_t = 1$ and an SVGP using an $\epsilon_0$ close k-DPP for selecting $\mathbf{Z}_t$, with probability at least $1 - \delta$, Assumptions 3 and 4 hold with parameters $c_t = \sqrt{\kappa_t}$, $\underline{a}_t = \frac{1}{\sqrt{1-\sqrt{3\kappa_t}}}$, $\bar{a}_t = \sqrt{1 + \sqrt{3\kappa_t}}$, and $\epsilon_t = \sqrt{C_1 m_t \delta_M}$, where $C_1$ is a constant specified in the appendix and $\kappa_t = \frac{2tB(m_t+1)\delta_{m_t}}{\tau\delta} + \frac{4tB\epsilon_0}{\tau\delta}$.*

*For S-GP-TS based on sampling rule (5) with $\alpha_t = 1$, Assumptions 3 and 4 hold with parameters $c_t = \sqrt{\kappa_t}$, $\underline{a}_t = \frac{1}{\sqrt{1-\sqrt{3\kappa_t}}}$, $\bar{a}_t = \sqrt{1 + \sqrt{3\kappa_t}}$, and $\epsilon_t = \sqrt{C_1 m_t \delta_M}$, where $C_1$ is the same constant as above and $\kappa_t = \frac{2tB\delta_{m_t}}{\tau}$.*

Note that our proposition requires extending the results of [26] in two non-trivial ways. First, the decoupled sampling rules introduce an additional error. Secondly, [26] built their convergence results on the assumption that the observation points $x_{t,b}$ are drawn from a prefixed distribution, which is not the case in S-GP-TS, where $x_{t,b}$ are selected according to an experimental design method. A detailed proof of Proposition 1 is provided in Appendix B.

## 5.3    Application of Regret Bounds to Matérn and SE Kernels

We now investigate the application of Theorem 1 to the Squared Exponential (SE) and Matérn kernels, widely used in practice [see, e.g., 17, 45]. In the case of a Matérn kernel with smoothness parameter $\nu > \frac{d}{2}$ it is known that $\lambda_j = O(j^{-\frac{2\nu+d}{d}})$ [46]. For the SE kernel, we have $\lambda_j = O(\exp(-j^{\frac{1}{d}}))$ [47, 48]. With these bounds on the spectrum of the kernels and the specific bounds on the maximal information gain [e.g., $\gamma_s \leq O(\log(s)^{d+1})$ for SE and $\gamma_s \leq O(s^{d/(2\nu+d)}\log(s))$ for Matérn, 49], Theorem 1 and Proposition 1 result in the following theorem.

**Theorem 2.** *Under Assumptions 1 and 2, with the algorithmic parameters, kernels and sampling rules specified in Table 1, S-GP-TS offers $R(T, B; f) = O(B\sqrt{\gamma_T \gamma_{TB} T \log(T)})$.*

With a batch size $B = 1$ Theorem 2 recovers the same regret bounds as the exact GP-TS [3]. We also note that for a fair comparison in terms of both the batch size and number of samples we should consider $T' = TB$ as the number of samples. In that case, our regret bound becomes $O(\sqrt{B\gamma_{T'/B}\gamma_{T'}T' \log(T'/B)})$, which scales at most with $\sqrt{B}$. That is $\sqrt{B}$ tighter than the trivial scaling with $B$.

In order to prove Theorem 2, the algorithmic parameters $M$ and $m_t$ must be selected large enough such that approximation parameters $\underline{a}, \bar{a}, c, \epsilon$ in Assumptions 3 and 4 are sufficiently small. Using the relation between the algorithmic parameters, the approximation parameters and $m_t$ provided by Proposition 1, the regret bound follows from Theorem 1. See Appendix B for a detailed proof.

The values of $M$ and $m_t$ required for Theorem 2 are summarized in Table 1. We also show the resulting computational cost of each sampling rule (as given by $O\left(B(M + m_T)N_T T + Bm_T^2 T^2\right)$), explicitly demonstrating the improvement of S-GP-TS over the $O(BN_T^3 T + B^3 T^4)$ computational cost of the vanilla GP-TS. Note that, for the Matérn kernel under sampling rule (4), $\nu$ is required to be sufficiently larger than $\frac{d}{2}$ in order for $m_t$ to grow slower than $t$.

| | | Inducing points (4) | Inducing features (5) |
|---|---|---|---|
| **Matérn** | Condition | $m_t \sim T^{\frac{2d}{2\nu-d}}, M \sim T^{\frac{(2\nu+d)d}{2(2\nu-d)\nu}}$ | $m_t \sim T^{\frac{d}{2\nu}}, M \sim T^{\frac{(2\nu+d)d}{4\nu^2}}$ |
| | Cost | $O\left(BN_T T^{\frac{4\nu^2+d^2}{2(2\nu-d)\nu}} + BT^2 \min\{T^{\frac{4d}{2\nu-d}}, T^2\}\right)$ | $O\left(BN_T T^{\frac{(2\nu+d)^2-2\nu d}{4\nu^2}} + BT^{\frac{2\nu+d}{\nu}}\right)$ |
| **SE** | Condition | $m_t, M \sim (\log(T))^d$ | $m_t, M \sim (\log(T))^d$ |
| | Cost | $O\left(BN_T T \log^d(T) + BT^2 \log^{2d}(T)\right)$ | $O\left(BN_T T \log^d(T) + BT^2 \log^{2d}(T)\right)$ |

Table 1: Conditions on the number of features $m_t$ and inducing variables $M_T$ required for Theorem 2, alongisde the resulting cost of each decoupled sampling method.

# 6 Experiments

We now provide an empirical evaluation of S-GP-TS. As [24] have already comprehensively demonstrated the practical advantage of decoupled sampling for problems with small optimization budgets, we focus here on scalability of S-GP-TS, and in particular a) its efficiency with large batch size, b) its ability to handle large data volumes. We first investigate a collection of classical synthetic problems for BO, before demonstrating S-GP-TS in a challenging real-world high-throughput molecular design considered by [13]. Our synthetic experiments focus on multi-modal problems with substantial observation noise, as these cannot be solved accurately with a small budget yet are still unsuitable for local, exhaustive, or deterministic optimization routines. Our implementation is provided as part of the open-source toolbox `trieste` [50] [2] and relies also on `gpflow` [51] and `gpflux` [52].

As is often the case, our regret-based analysis applies to a version of the algorithm that is slightly different to a practically viable BO method. Rather than focusing on recreating our algorithm exactly in the very limited settings (e..g a 1-d RBF kernel for which we can calculate eigen-features exactly) that are of little interest to the BO community, we instead choose to demonstrate the practical strength and unprecedented scalability of S-GP-TS by investigating an implementation that could be used by practitioners. The resulting algorithms demonstrated in this section are still well-aligned with our work through their use of sparse GP surrogate models and decoupled Thompson sampling.

## 6.1 Synthetic Benchmarks

We first consider two toy problems: Hartmann (6 dim, moderately multi-modal) with a large additive noise and Shekel (4 dim, highly multi-modal) with moderate noise, see Appendix C for the full description. Our SVGP models use inducing points and a Matérn kernel with smoothness parameter

[2] https://github.com/secondmind-labs/trieste

$\nu = 2.5$. As eigenfunctions for this kernel are limited to small dimensions [39], we implement decoupled TS using the easily accessible random Fourier Features (RFF). Note that [24] have shown decoupled sampling to significantly alleviate the *variance starvation* phenomenon (underestimating the variance of points far from the observations [16, 43]) that typically hampers the efficacy of RFFs. We use $M = 1000$ features and maximise each sample as in (3) using L-BFGS-B, starting from the best point among a large sample.

As sampling inducing points from a k-DPP is prohibitively costly for the repeated model fitting required by BO loops, we use the the greedy variance selection method of [26] which is $\epsilon_0$ *close* to k-DPP and has been shown to outperform optimisation of inducing points in practice. We also consider the practical alternative of choosing inducing points chosen by a k-means clustering of the observations. As the optimisation progresses, observations are likely to be concentrated in the optimal regions, so clustering would result in somehow "targeted" inducing points for BO. In order to control the computational overhead of S-GP-TS and to allow an efficient computational implementation (i.e. avoiding Tensorflow recompilation issues), we use a fixed number $m_t$ of points, set to either 250 or 500. Similarly, we set the covariance scaling parameter $\alpha_t = 1$ to avoid having dynamic tunable parameters, like those that plague UCB-based approaches.

For each experiment, we run $t = 50$ steps of S-GP-TS with $B = 100$ (i.e. 5,000 total observations). For baselines, we compare against $t = 750$ steps of standard sequential non-batch BO routines with an exact GP model: Expected Improvement [EI, 53], Augmented Expected Improvement [AEI, 54], and an extension of Max-value Entropy search suitable for noisy observations [GIBBON, 10]. Due to the large number of steps, we only consider low-cost but high-performance acquisitions, following the cost-benefit analysis of [10], and exclude the popular knowledge gradient [9] or classical entropy search [55, 40]. Popular existing batch acquisition functions do not scale to batches as large as $B = 100$, however, we present their performance on smaller batches across additional experiments in Appendix C. We report simple regret of the current believed best solution (maximizer of the current model mean) across the previously queried data points. All results are averaged over 30 runs and reported as a function of either the number of function evaluations ($tB$ for S-GP-TS and $t$ for the baselines), or the number of BO iterations, in Figure 1.

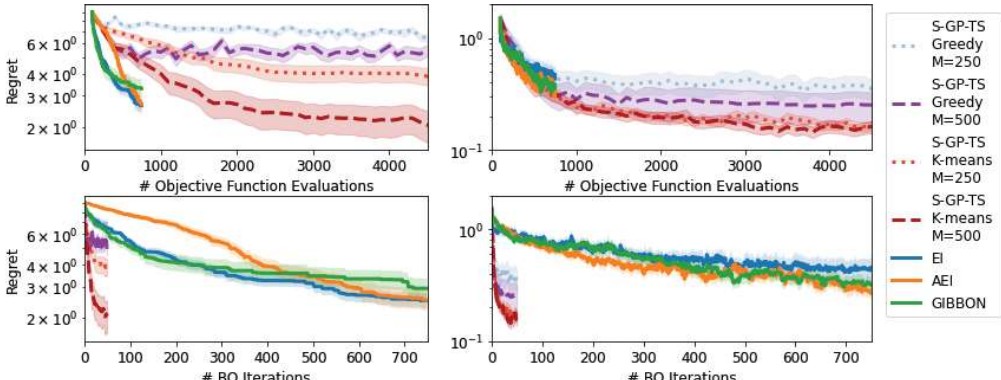

Figure 1: Simple regret on Shekel (4D, left) and Hartmann (6D, right). When considering regret with respect to the total number of objective function evaluations $tB$ (top panels, purely sequential setting), all S-GP-TS methods are initially less efficient (Shekel) or match the performance (Hartmann) of the best baselines, however the best S-GP-TS approach is able to efficiently allocate its additional budget to achieve lower final regret. When considering regret with respect to the BO iteration (bottom panels, idealised parallel setting), S-GP-TS achieves low regret in a fraction of the iterations required by the standard BO routines.

The fact that S-GP-TS is able to find solutions on both benchmarks with substantially improved regret than found by standard BO, provides strong evidence that S-GP-TS is effectively leveraging parallel resources. Moreover, as these higher-quality solutions were only found after large number of total evaluations, Figure 1 also highlights the necessity for BO routines, like S-GP-TS, that can handle these larger (heavily parallelized) optimization budgets. We reiterate that the existing BO baselines cannot handle as many evaluations as S-GP-TS, becoming prohibitively slow once we surpass 750 data-points). When considering the regret achieved per individual function evaluation, we typically expect batch routines to be less efficient than purely sequential BO routines. However, in the case of the Hartmann function (the benchmark with the largest observation noise), we see that our best

S-GP-TS exactly matches (before going on to exceed) the performance of the sequential routines, suggesting that S-GP-TS is a particularly effective optimizer for functions with significant levels of observation noise.

Note that the performance of S-GP-TS is sensitive to its chosen inducing points, with k-means providing the most effective routines. On Hartmann, 250 inducing points is sufficient to deliver good performances, while on Shekel, which is much more multimodal, using a larger number is critical.

### 6.2  High-throughput Molecular Search

Finally, we investigate the performance of S-GP-TS with respect to an established baseline for high-throughput molecular screening. Although molecular search has been tackled many times with BO [56, 22, 57], only the approach of [13] - standard (non-decoupled) TS over a Bayesian neural network (BNN-TS) - is truly scalable. We now recreate the largest experiment considered by [13], where the objective is to uncover the top $10\%$ of molecules in terms of power conversion efficiency among a library of 2.3 million candidate from the Harvard Clean Energy Project [58]. Molecules are encoded as Morgan circular fingerprints of Bond radius 3 (i.e. 512-dimensional bit vectors, see [59]).

As the standard GP kernels considered above are not suitable for sparse and high-dimensional molecule inputs [60], we instead build our SVGP with a zeroth order ArcCosine kernel [61], chosen due to its strong empirical performance under sparsity and as it permits a random decomposition that can be exploited to perform decoupled TS. In particular, we use the $M$-feature decomposition investigated by [62] of

$$k_{arc}(\mathbf{x}, \mathbf{x}') = 2 \int d\mathbf{w} \frac{\mathrm{e}^{-\frac{\|\mathbf{w}\|^2}{2}}}{(2\pi)^{d/2}} \Theta(\mathbf{w}^T\mathbf{x})\Theta(\mathbf{w}^T\mathbf{x}') \approx \frac{2}{M} \sum_{j=1}^{M} \Theta(\mathbf{w}_j^T\mathbf{x})\Theta(\mathbf{w}_j^T\mathbf{x}'),$$

where $\Theta(.)$ is the Heaviside step function and $\mathbf{w}_j \sim \mathcal{N}(0, I)$.

In our experiments, we use $M = 1\,000$ random features and, to avoid memory issues, we compute our GP samples over a random subset of $100\,000$ molecules (renewed at each sample). We run S-GP-TS twice, once with $m_t = 500$ and once with $2000$ inducing points. We chose inducing points as uniform samples from the already evaluated molecules (for each model step), as preliminary experiments showed that neither the k-means nor greedy selection routines discussed above were effective when applied to sparse and high-dimensional molecular fingerprint inputs.

Following [13], we report the recall (fraction of the top $10\%$ of molecules so far chosen by the BO loop) for S-GP-TS, along with the performance of BNN-TS, a greedy BNN (that queries the $B$ maximizers of the BNN's posterior mean), and a random search baseline (all taken from [13]). All routines (including our S-GP-TS) are ran for $t = 250$ successive batches of $B = 500$ molecules. Figure 2 shows that S-GP-TS is able to perform effective batch optimization over very large optimization budgets (120,000 total evaluations) and, when using $m = 2000$ or even just $m = 500$ inducing points, S-GP-TS matches the performance of [13]'s BNN-based TS and greedy sampling approaches, respectively. Note that due to the high computational demands of this experiment, we report just a single replication of S-GP-TS (a limitation also of [13]'s results). However, we stress that an additional realization of the $m = 500$ experiment returned indistinguishable results.

## 7  Discussion

We have shown that S-GP-TS enjoys the same regret order as exact GP-TS but with a greatly reduced $O(N_t M)$ computation per step $t$, compared to the $O(N_t^3)$ cost of the standard sampling. However, the discretization size $N_t$ is exponential in the dimension $d$ of the search space and so remains a limiting computational factor when optimizing over high dimensional search spaces. Hence, while S-GP-TS with decoupled sampling rule allows orders of magnitude larger optimization budgets compared to vanilla GP-TS, it still suffers from the *curse of dimensionality*. Intuitively, this seems inevitable due to NP-Hardness of non-convex optimization problems [see, e.g., 63] as required to find the maximizer of the GP-UCB acquisition function [see, e.g., 30], or even in the application of UCB to linear bandits [64]. In particular, the computational cost of the *state-of-the-art* adaptive sketching method for implementing GP-UCB [30] was reported as $O(N_T d_{\mathrm{eff}}^2)$ where $d_{\mathrm{eff}}$, referred to as the effective dimension of the problem, is upper bounded by $\gamma_T$.

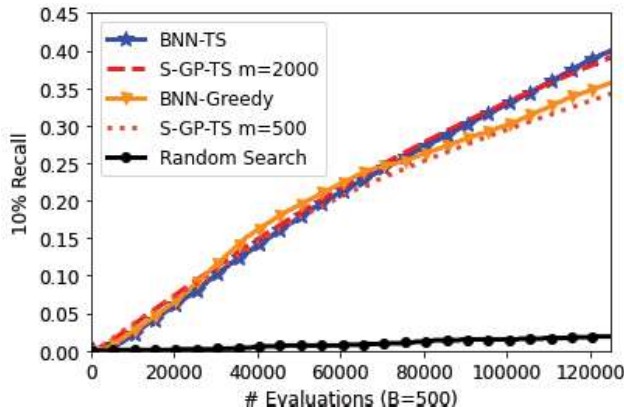

Figure 2: Proportion of the top 10% of molecules found by each of the search routines. S-GP-TS is able to process substantial data volumes and effectively allocates large batches, matching the performance of the well-established BNN baselines.

An important practical consideration when using S-GP-TS in practice is how to choose its inducing points. The performance improvement provided by choosing inducing points by k-means rather than greedy variance selection, as demonstrated in our experiments, raises the possibility that BO-specific routines for choosing inducing points could allow even better performance. This is an important avenue for future work.

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
