# A Complements on SVGPs

As discussed in Section 3, SVGPs approximate the posterior of exact GPs through either a set of inducing points $\mathbf{Z}_t \triangleq \{z_1, ..., z_m\}$ or through a set of inducing features $\phi_m(x) \triangleq \{\phi_1(x), ..., \phi_m(x)\}$. The resulting inducing features, defined as $u_{t,i} = \hat{f}(z_{t,i})$ (for inducing points) or $u_{t,i} = \int_{\mathcal{X}} \hat{f}(x)\phi_i(x)dx$ (for inducing features), are assumed to follow a prior Gaussian density $q_t(\mathbf{u}_t) = \mathcal{N}(\mathbf{m}_t, \mathbf{S}_t)$. We now discuss how to set these variational parameters $\mathbf{m}_t$ and $\mathbf{S}_t$ for a given dataset.

For SVGPS, the posterior mean and covariance is given in closed form as

$$\mu_t^{(s)}(x) = k_{\mathbf{Z}_t,x}^{\mathrm{T}} K_{\mathbf{Z}_t,\mathbf{Z}_t}^{-1}\mathbf{m}_t \quad k_t^{(s)}(x,x') = k(x,x') + k_{\mathbf{Z}_t,x}^{\mathrm{T}} K_{\mathbf{Z}_t,\mathbf{Z}_t}^{-1}(\mathbf{S}_t - K_{\mathbf{Z}_t,\mathbf{Z}_t}) K_{\mathbf{Z}_t,\mathbf{Z}_t}^{-1} k_{\mathbf{Z}_t,x'}$$

and

$$\mu_t^{(s)}(x) = \phi_{m_t}^{\mathrm{T}}(x)\mathbf{m}_t \qquad k_t^{(s)}(x,x') = k(x,x') + \phi_{m_t}^{\mathrm{T}}(x)(\mathbf{S}_t - \Lambda_{m_t})\phi_{m_t}(x'),$$

for the inducing point and inducing features representations, respectively. See Section 3 or [26] for more details. However, the marginal likelihood for these models is intractable, and so, as is common practice in variational inference methods, we set of our variational parameters (as-well as the SVGP's kernel parameters) to maximize instead the tractable Evidence-based Lower BOund (ELBO).

For inducing point SVGPS, the ELBO can be written as

$$\mathrm{ELBO}(t) \quad = \quad -\frac{1}{2}\mathbf{y}_t^{\mathrm{T}}(Q_t + \tau\mathbf{I}_t)^{-1}\mathbf{y}_t - \frac{1}{2}\log|Q_t + \tau\mathbf{I}_t| - \frac{t}{2}\log(2\pi) - \frac{\theta_t}{2\tau},$$

where $Q_t = K_{\mathbf{Z}_t,\mathbf{X}_t}^{\mathrm{T}} K_{\mathbf{Z}_t,\mathbf{Z}_t}^{-1} K_{\mathbf{Z}_t,\mathbf{X}_t}, K_{\mathbf{Z}_t,\mathbf{X}_t} = [k_{z_i,x_j}]_{i,j}, i = 1, \ldots, m_t, j = 1, \ldots, t, \mathbf{I}_t$ is the $t \times t$ identity matrix and $\theta_t = \mathrm{Tr}(K_{\mathbf{X}_t,\mathbf{X}_t} - Q_t)$. See [32] for a full derivation.

For inducing feature SVGP, the expression of ELBO is the same but with $Q_t = K_{\phi_{m_t},\mathbf{X}_t}^{\mathrm{T}} \Lambda_{m_t}^{-1} K_{\phi_{m_t},\mathbf{X}_t}, K_{\phi_{m_t},\mathbf{X}_t} = [\lambda_i\phi_i(x_j)]_{i,j}, i = 1, \ldots, m_t, j = 1, \ldots, t$.

To optimize the ELBO in practice, [32] proposed a numerical solution allowing for mini-batching [see also 26] and the use of stochastic gradient descent algorithms such as Adam [65]. In addition, [20] provides an explicit solution for the convex optimization problem of finding $(\mathbf{m}_t, \mathbf{S}_t)$, allowing more involved alternate optimization schemes.

# B Detailed Proofs

In this section, we provide detailed proofs for Theorem 1, Lemma 1, Proposition 1 and Theorem 2, in order.

## B.1 Proof of Theorem 1

Before presenting the proof of Theorem 1, we first overview the regret bound for vanilla GP-TS [3, Theorem 4].

*The Existing Regret Bound for Vanilla GP-TS.* [3] proved that, with probability at least $1 - \delta$, $|f(x) - \mu_t(x)| \leq u_t\sigma_t(x)$, where $u_t = \left(B + R\sqrt{2(\gamma_t + 1 + \log(1/\delta))}\right)$ and $\gamma_t$ is the maximal information gain. Based on this concentration inequality, [3] showed that the regret of GP-TS scales with the cumulative uncertainty at the observation points measured by the standard deviation: $O(\sum_{t=1}^{T} u_t\sigma_{t-1}(x_t))$. Furthermore, [29] showed that $\sum_{i=1}^{t} \sigma_{i-1}^2(x_i) \leq \gamma_t$. Using this result and applying Cauchy-Schwarz inequality to $O(\sum_{t=1}^{T} u_t\sigma_{t-1}(x_t))$, [3] proved that $R(T,1;f) = O\left(\gamma_T\sqrt{T\log(T)}\right)$, for vanilla GP-TS.

$\square$

We build on the analysis of GP-TS in [3] to prove the regret bounds for S-GP-TS. We stress that despite some similarities in the proof, the analysis of standard GP-TS does not extend to S-GP-TS. This proof characterizes the behavior of the upper bound on regret in terms of the approximation

constants, namely $\underline{a}, \bar{a}, c$ and $\epsilon$. A notable difference is that the additive approximation error in the posterior standard deviation ($\epsilon_t$) can cause under-exploration which is an issue the analysis of exact GP-TS cannot address. In addition, we account for the effect of batch sampling on the regret bounds.

We first focus on the instantaneous regret at each time $t$ within the discrete set, $f(\mathbf{x}^{*(t)}) - f(x_{t,b})$. Recall $\mathbf{x}^{*(t)} \triangleq \operatorname{argmin}_{x' \in D_t} \|x^* - x'\|$ from Assumption 5. It is then easy to upper bound the cumulative regret by the cumulative value of $f(\mathbf{x}^{*(t)}) - f(x_{t,b}) + \frac{1}{t^2}$ as our discretization ensures that $f(x^*) - f(\mathbf{x}^{*(t)}) \leq \frac{1}{t^2}$. For upper bounds on instantaneous regret, we start with concentration of GP samples $\tilde{f}_{t,b}$ around their predicted values and the concentration of the prediction around the true objective function. We then consider the anti-concentration around the optimum point. The necessary anti-concentration may fail due to approximation error in the standard deviation around the optimum point. We thus consider two cases of low and sufficiently high standard deviation at $\mathbf{x}^{*(t)}$ separately. While a low standard deviation implies good prediction at $\mathbf{x}^{*(t)}$, a sufficiently high standard deviation guarantees sufficient exploration. We use these results to upper bound the instantaneous regret at each time $t$ with uncertainties measured by the standard deviation.

**Concentration events $\mathcal{E}_t$ and $\tilde{\mathcal{E}}_t$:**

**Define** $\mathcal{E}_t$ as the event that at time $t$, for all $x \in D_t$, $|f(x) - \tilde{\mu}_{t-1}(x)| \leq \frac{1}{2}\alpha_t(\tilde{\sigma}_{t-1}(x) + \epsilon_t)$. Recall $\alpha_t = 2\tilde{u}_t(1/(t^2))$. Applying lemma 1, we have $\Pr[\mathcal{E}_t] \geq 1 - \frac{1}{t^2}$.

**Define** $\tilde{\mathcal{E}}_t$ as the event that for all $x \in D_t$, and for all $b \in [B]$, $|\tilde{f}_{t,b}(x) - \tilde{\mu}_{t-1}(x)| \leq \alpha_t b_t \tilde{\sigma}_{t-1}(x)$ where $b_t = \sqrt{2\ln(BN_t t^2)}$. We have $\Pr[\tilde{\mathcal{E}}_t] \geq 1 - \frac{1}{t^2}$.

*Proof.* For a fixed $x \in D_t$, and a fixed $b \in [B]$,

$$\Pr\left[|\tilde{f}_{t,b}(x) - \tilde{\mu}_{t-1}(x)| > \alpha_t b_t \tilde{\sigma}_{t-1}(x)\right] < \exp(-\frac{b_t^2}{2}) = \frac{1}{BN_t t^2}.$$

The inequality holds because of the following bound on the CDF of a normal random variable $1 - \text{CDF}_{\mathcal{N}(0,1)}(c) \leq \frac{1}{2}\exp(-\frac{c^2}{2})$ and the observation that $\frac{\tilde{f}_{t,b}(x) - \tilde{\mu}_{t-1}(x)}{\alpha_t \tilde{\sigma}_{t-1}(x)}$ has a normal distribution. Applying a union bound we get $\Pr[\bar{\tilde{\mathcal{E}}}_t] \leq \frac{1}{t^2}$ which gives us the bound on probability of $\tilde{\mathcal{E}}_t$. $\qquad \square$

We thus proved $\mathcal{E}_t$ and $\tilde{\mathcal{E}}_t$ are high probability events. This will facilitate the proof by conditioning on $\mathcal{E}_t$ and $\tilde{\mathcal{E}}_t$. Also notice that when both $\mathcal{E}_t$ and $\tilde{\mathcal{E}}_t$ hold true, we have, for all $x \in D_t$, and for all $b \in [B]$,

$$|\tilde{f}_{t,b}(x) - f(x)| \leq \beta_t \tilde{\sigma}_{t-1}(x) + \frac{1}{2}\alpha_t \epsilon_t \tag{7}$$

where $\beta_t = \alpha_t(b_t + \frac{1}{2})$.

**Anti Concentration Bounds.** It is standard in the analysis of TS methods to prove sufficient exploration using an anti-concentration bound. That establishes a lower bound on the probability of a sample being sufficiently large (so that the corresponding point is likely to be selected by TS rule). For this purpose, we use the following bound on the CDF of a normal distribution: $1 - \text{CDF}_{\mathcal{N}(0,1)}(c) \geq \frac{\exp(-c^2)}{4c\sqrt{\pi}}$. The underestimation of the posterior standard deviation at the optimum point however might result in an under exploration. On the other hand, a low standard deviation at the optimum point implies a low prediction error. We use this observation in our regret analysis by considering the two cases separately. Specifically, the regret $f(\mathbf{x}^{*(t)}) - f(x_{t,b})$ at each time $t$ for each sample $b$ is bounded differently under the conditions: I. $\tilde{\sigma}_{t-1}(\mathbf{x}^{*(t)}) \leq \epsilon_t$ and II. $\tilde{\sigma}_{t-1}(\mathbf{x}^{*(t)}) > \epsilon_t$.

**Under Condition** I ($\tilde{\sigma}_{t-1}(\mathbf{x}^{*(t)}) \leq \epsilon_t$), when both $\mathcal{E}_t$ and $\tilde{\mathcal{E}}_t$ hold true, we have

$$
\begin{aligned}
f(\mathbf{x}^{*(t)}) &- f(x_{t,b}) \\
&\leq \quad \tilde{f}_{t,b}(\mathbf{x}^{*(t)}) + \beta_t \tilde{\sigma}_{t-1}(\mathbf{x}^{*(t)}) + \frac{1}{2}\alpha_t \epsilon_t \\
&\qquad - \tilde{f}_t(x_{t,b}) + \beta_t \tilde{\sigma}_{t-1}(x_{t,b}) + \frac{1}{2}\alpha_t \epsilon_t \qquad\qquad \text{by (7)},\\
&\leq \quad \beta_t \tilde{\sigma}_{t-1}(\mathbf{x}^{*(t)}) + \beta_t \tilde{\sigma}_{t-1}(x_{t,b}) + \alpha_t \epsilon_t \quad \text{by the selection rule of TS,} \qquad (8)\\
&\leq \quad \beta_t \tilde{\sigma}_{t-1}(x_{t,b}) + (\beta_t + \alpha_t)\epsilon_t \qquad\qquad\qquad \text{by Condition I.}
\end{aligned}
$$

that upper bounds the instantaneous regret at time $t$ by a factor of approximate standard deviation up to an additive term caused by approximation error. Since $f(\mathbf{x}^{*(t)}) - f(x_{t,b}) \leq 2B$, under Condition I,

$$
\mathbb{E}[f(\mathbf{x}^{*(t)}) - f(x_{t,b})] \leq \beta_t \tilde{\sigma}_{t-1}(x_{t,b}) + (\beta_t + \alpha_t)\epsilon_t + \frac{4B}{t^2}. \tag{9}
$$

where the inequality holds by $\Pr[\bar{\mathcal{E}}_t \text{ or } \bar{\tilde{\mathcal{E}}}_t] \leq \frac{2}{t^2}$.

**Under Condition** II ($\tilde{\sigma}_{t-1}(\mathbf{x}^{*(t)}) > \epsilon_t$), we can show sufficient exploration by anti-concentration at the optimum point. In particular under Condition II, if $\mathcal{E}_t$ holds true, we have

$$
\Pr[\tilde{f}_{t,b}(\mathbf{x}^{*(t)}) > f(\mathbf{x}^{*(t)})] \geq p, \tag{10}
$$

where $p = \frac{1}{4\sqrt{\pi}}$.

*Proof.* Applying the anti-concentration of a normal distribution

$$
\begin{aligned}
\Pr[\tilde{f}_{t,b}(\mathbf{x}^{*(t)}) > f(\mathbf{x}^{*(t)})] &= \Pr\left[ \frac{\tilde{f}_{t,b}(\mathbf{x}^{*(t)}) - \tilde{\mu}_{t-1}(\mathbf{x}^{*(t)})}{\alpha_t \tilde{\sigma}_{t-1}(\mathbf{x}^{*(t)})} > \frac{f(\mathbf{x}^{*(t)}) - \tilde{\mu}_{t-1}(\mathbf{x}^{*(t)})}{\alpha_t \tilde{\sigma}_{t-1}(\mathbf{x}^{*(t)})} \right] \\
&\geq p.
\end{aligned}
$$

As a result of the observation that the right hand side of the inequality inside the probability argument is upper bounded by 1:

$$
\begin{aligned}
\frac{f(\mathbf{x}^{*(t)}) - \tilde{\mu}_{t-1}(\mathbf{x}^{*(t)})}{\alpha_t \tilde{\sigma}_{t-1}(\mathbf{x}^{*(t)})} &\leq \frac{\frac{1}{2}\alpha_t \tilde{\sigma}_{t-1}(\mathbf{x}^{*(t)}) + \frac{1}{2}\alpha_t \epsilon_t}{\alpha_t \tilde{\sigma}_{t-1}(\mathbf{x}^{*(t)})} \quad \text{By } \mathcal{E}_t \\
&\leq 1. \qquad\qquad\qquad\qquad\qquad \text{By Condition II} \quad \square
\end{aligned}
$$

**Sufficiently Explored Points.** Let $\mathcal{S}_t$ denote the set of sufficiently explored points which are unlikely to be selected by S-GP-TS if $\tilde{f}_{t,b}(\mathbf{x}^{*(t)})$ is higher than $f(\mathbf{x}^{*(t)})$. Specifically, we use the notation

$$
\mathcal{S}_t = \{x \in D_t : f(x) + \beta_t \tilde{\sigma}_{t-1}(x) + \frac{1}{2}\alpha_t \epsilon_t \leq f(\mathbf{x}^{*(t)})\}. \tag{11}
$$

Recall $\beta_t = \alpha_t(b_t + \frac{1}{2})$. In addition, we define

$$
\bar{x}_t = \operatorname{argmin}_{x \in D_t \setminus \mathcal{S}_t} \tilde{\sigma}_{t-1}(x). \tag{12}
$$

We showed in equation (8) that the instantaneous regret can be upper bounded by the sum of standard deviations at $x_{t,b}$ and $\mathbf{x}^{*(t)}$. The standard method based on information gain can be used to bound the cumulative standard deviations at $x_{t,b}$. This is not sufficient however because the cumulative standard deviations at $\mathbf{x}^{*(t)}$ do not converge unless there is sufficient exploration around $x^*$. To address this, we use $\bar{x}_t$ as an intermediary to be able to upper bound the instantaneous regret by a factor of $\tilde{\sigma}_{t-1}(x_{t,b})$ through the following lemma.

**Lemma 2.** *Under Condition II, for $t \geq \sqrt{\frac{2}{p}}$, if $\mathcal{E}_t$ holds true*

$$
\tilde{\sigma}_{t-1}(\bar{x}_t) \leq \frac{2}{p}\mathbb{E}[\tilde{\sigma}_{t-1}(x_{t,b})], \tag{13}
$$

*where the expectation is taken with respect to the randomness in the sample $\tilde{f}_{t,b}$.*

*Proof of Lemma 2.* First notice that when both $\mathcal{E}_t$ and $\tilde{\mathcal{E}}_t$ hold true, for all $x \in \mathcal{S}_t$

$$
\begin{aligned}
\tilde{f}_{t,b}(x) &\leq f(x) + \beta_t \tilde{\sigma}_{t-1}(x) + (\alpha_t - 1)\epsilon_t && \text{by (7)} \\
&\leq f(\mathbf{x}^{*(t)}), && \text{by definition of } \mathcal{S}_t.
\end{aligned}
\tag{14}
$$

Also, if $\tilde{f}_{t,b}(\mathbf{x}^{*(t)}) > \tilde{f}_{t,b}(x), \forall x \in S_t$, the rule of selection in TS ($x_{t,b} = \operatorname{argmax}_{x \in \mathcal{X}} \tilde{f}_{t,b}(x)$) ensures $x_{t,b} \in D_t \setminus \mathcal{S}_t$. So we have

$$
\begin{aligned}
\Pr[x_{t,b} \in D_t \setminus \mathcal{S}_t] &\geq \Pr[\tilde{f}_{t,b}(\mathbf{x}^{*(t)}) > \tilde{f}_{t,b}(x), \forall x \in S_t] \\
&\geq \Pr[\tilde{f}_{t,b}(\mathbf{x}^{*(t)}) > \tilde{f}_{t,b}(x), \forall x \in S_t, \tilde{\mathcal{E}}_t] - \Pr[\bar{\tilde{\mathcal{E}}}_t] \\
&\geq \Pr[\tilde{f}_{t,b}(\mathbf{x}^{*(t)}) > f(\mathbf{x}^{*(t)})] - \Pr[\bar{\tilde{\mathcal{E}}}_t] && \text{by (14)} \\
&\geq p - \frac{1}{t^2} && \text{by (10)} \\
&\geq \frac{p}{2}, && \text{for } t \geq \sqrt{2/p}.
\end{aligned}
$$

Finally, we have

$$
\begin{aligned}
\mathbb{E}[\tilde{\sigma}_{t-1}(x_{t,b})] &\geq \mathbb{E}\left[\tilde{\sigma}_{t-1}(x_{t,b}) \,\middle|\, x_{t,b} \in D_t \setminus S_t\right] \Pr[x_{t,b} \in D_t \setminus S_t] \\
&\geq \frac{p\tilde{\sigma}_{t-1}(\bar{x}_t)}{2},
\end{aligned}
\tag{15}
$$

where the expectation is taken with respect to the randomness in the sample $\tilde{f}_{t,b}$ at time $t$. $\qquad \square$

Now we are ready to bound the simple regret under Condition II using $\bar{x}_t$ as an intermediary. Under Condition II, when both $\mathcal{E}_t$ and $\tilde{\mathcal{E}}_t$ hold true,

$$
\begin{aligned}
f(\mathbf{x}^{*(t)}) - f(x_{t,b}) &= f(\mathbf{x}^{*(t)}) - f(\bar{x}_t) + f(\bar{x}_t) - f(x_{t,b}) \\
&\leq \beta_t \tilde{\sigma}_{t-1}(\bar{x}_t) + \frac{1}{2}\alpha_t \epsilon_t + f(\bar{x}_t) - f(x_{t,b}) && \text{by definition of } \mathcal{S}_t \\
&\leq \beta_t \tilde{\sigma}_{t-1}(\bar{x}_t) + \frac{1}{2}\alpha_t \epsilon_t \\
&\qquad + \tilde{f}_{t,b}(\bar{x}_t) + \beta_t \tilde{\sigma}_{t-1}(\bar{x}_t) - \tilde{f}_{t,b}(x_{t,b}) + \beta_t \tilde{\sigma}_{t-1}(x_{t,b}) + \alpha_t \epsilon_t && \text{by (7)} \\
&\leq \beta_t(2\tilde{\sigma}_{t-1}(\bar{x}_t) + \tilde{\sigma}_{t-1}(x_{t,b})) + \frac{3}{2}\alpha_t \epsilon_t, && \text{by the rule of selection in TS.}
\end{aligned}
$$

Thus, since $f(x^*) - f(x_{t,b}) \leq 2B$, under Condition II, for $t \geq \sqrt{\frac{2}{p}}$

$$
\mathbb{E}[f(\mathbf{x}^{*(t)}) - f(x_{t,b})] \leq \frac{(4+p)\beta_t}{p}\mathbb{E}[\tilde{\sigma}_{t-1}(x_{t,b})] + \frac{3}{2}\alpha_t \epsilon_t + \frac{4B}{t^2}
\tag{16}
$$

where we used Lemma 2 and $\Pr[\bar{\mathcal{E}}_t \text{ or } \bar{\tilde{\mathcal{E}}}_t] \leq \frac{2}{t^2}$.

**Upper bound on regret.** From the upper bounds on instantaneous regret under Condition I and Condition II we conclude that, for $t \geq \sqrt{\frac{2}{p}}$

$$
\begin{aligned}
\mathbb{E}[f(\mathbf{x}^{*(t)}) - f(x_{t,b})] &\leq \max\left\{ \beta_t \tilde{\sigma}_{t-1}(x_{t,b}) + (\beta_t + \alpha_t)\epsilon_t + \frac{4B}{t^2}, \right. \\
&\qquad\qquad \left. \frac{(4+p)\beta_t}{p}\mathbb{E}[\tilde{\sigma}_{t-1}(x_{t,b})] + \frac{3}{2}\alpha_t \epsilon_t + \frac{4B}{t^2} \right\} \\
&\leq \frac{(4+p)\beta_t}{p}\mathbb{E}[\tilde{\sigma}_{t-1}(x_{t,b})] + (\beta_t + \alpha_t)\epsilon_t + \frac{4B}{t^2}.
\end{aligned}
\tag{17}
$$

We can now upper bound the cumulative regret. Noticing $\lceil \sqrt{\frac{2}{p}} \rceil = 4$.

$$
\begin{aligned}
R(T, B; f) &= \sum_{t=1}^{T} \sum_{b=1}^{B} \mathbb{E}[f(x^*) - f(x_{t,b})] \\
&= \sum_{t=1}^{4} \sum_{b=1}^{B} \mathbb{E}[f(x^*) - f(x_{t,b})] + \sum_{t=5}^{T} \sum_{b=1}^{B} \mathbb{E}[f(x^*) - f(x_{t,b})] \\
&\leq 8B\mathcal{B} + \sum_{t=5}^{T} \left( \mathbb{E}[f(\mathbf{x}^{*(t)}) - f(x_{t,b})] + \frac{1}{t^2} \right) \\
&\leq 8B\mathcal{B} + \sum_{t=5}^{T} \sum_{b=1}^{B} \left( \frac{(4+p)\beta_t}{p} \mathbb{E}[\tilde{\sigma}_{t-1}(x_{t,b})] + (\beta_t + \alpha_t)\epsilon_t + \frac{4\mathcal{B}+1}{t^2} \right) \\
&\leq 8B\mathcal{B} + \frac{\pi^2 B(4\mathcal{B}+1)}{6} + \frac{(4+p)\beta_T}{p} \sum_{t=1}^{T} \sum_{b=1}^{B} \mathbb{E}[\tilde{\sigma}_{t-1}(x_{t,b})] + (\beta_T + \alpha_T) \sum_{t=1}^{T} \sum_{b=1}^{B} \epsilon_t \\
&\leq 15B\mathcal{B} + 2B + 30\beta_T \sum_{t=1}^{T} \sum_{b=1}^{B} (\bar{a}\mathbb{E}[\sigma_{t-1}(x_{t,b})] + \epsilon_t) + (\beta_T + \alpha_T)\epsilon TB \\
&\leq 15B\mathcal{B} + 2B + 30\bar{a}\beta_T \sum_{t=1}^{T} \sum_{b=1}^{B} \mathbb{E}[\sigma_{t-1}(x_{t,b})] + 30\beta_T \epsilon TB + (\beta_T + \alpha_T)\epsilon TB \\
&\leq 15B\mathcal{B} + 2B + 30\bar{a}\beta_T \sum_{t=1}^{T} \sum_{b=1}^{B} \mathbb{E}[\sigma_{t-1}(x_{t,b})] + (31\beta_T + \alpha_T)\epsilon TB.
\end{aligned}
$$

We simplified the expressions by $\frac{4+p}{p} \leq 30$, $\frac{4\pi^2}{6} \leq 7$ and $\frac{\pi^2}{6} \leq 2$.

We now use a technique based on information gain to upper bound $\sum_{t=1}^{T} \sum_{b=1}^{B} \mathbb{E}[\sigma_{t-1}(x_{t,b})]$ as formalized in the following lemma.

**Lemma 3.** *For all batch observation sequences $\{x_{t,b}\}_{t\in[T],b\in[B]}$, we have*

$$
\sum_{t=1}^{T} \sum_{b=1}^{B} \sigma_{t-1}(x_{t,b}) \leq B\sqrt{\frac{2T\gamma_T}{\log(1 + \frac{1}{\tau})}} \tag{18}
$$

*Proof of Lemma 3.* Without loss of generality assume that at each time instance $t = 1, 2, \ldots, T$, the batch observations are ordered such that $\sigma_{t-1}(x_{t,1}) \geq \sigma_{t-1}(x_{t,b})$, for all $b \in [B]$. We thus have

$$
\sum_{t=1}^{T} \sum_{b=1}^{B} \sigma_{t-1}(x_{t,b}) \leq B \sum_{t=1}^{T} \sigma_{t-1}(x_{t,1}). \tag{19}
$$

For the sequence of observations $\{x_{t,1}\}_{t=1}^{T}$, define the conditional posterior mean and variance

$$
\begin{aligned}
\bar{\mu}_t(x) &= \mathbb{E}[\hat{f}(x)|\{x_{s,1}\}_{s=1}^{t}] \\
\bar{\sigma}_t^2(x) &= \mathbb{E}[(\hat{f}(x) - \bar{\mu}_t(x))^2|\{x_{s,1}\}_{s=1}^{t}].
\end{aligned}
$$

By the expression of posterior variance of multivariate Gaussian random variables and by positive definiteness of the covariance matrix, we know that conditioning on a larger set reduces the posterior variance. Thus $\bar{\sigma}_t(x) \geq \sigma_t(x)$. Notice that $\sigma_t(x)$ is the posterior variance conditioned on full batches of the observations while $\bar{\sigma}_t(x)$ is the posterior variance conditioned on only the first observation at each batch. We thus have

$$
\sum_{t=1}^{T} \sigma_{t-1}(x_{t,1}) \leq \sum_{t=1}^{T} \bar{\sigma}_{t-1}(x_{t,1}) \tag{20}
$$

We can now follow the standard steps in bounding the cumulative standard deviation in the non-batch setting. In particular using Cauchy-Schwarz inequality, we have

$$\sum_{t=1}^{T} \bar{\sigma}_{t-1}(x_{t,1}) \leq \sqrt{T \sum_{t=1}^{T} \bar{\sigma}_{t-1}^2(x_{t,1})}. \tag{21}$$

In addition, [29] showed that

$$\sum_{t=1}^{T} \bar{\sigma}_{t-1}^2(x_{t,1}) \leq \frac{2\gamma_T}{\log(1 + \frac{1}{\tau})}. \tag{22}$$

Combining (19), (20), (21) and (22), we arrive at the lemma.

$\square$

We thus have

$$R(T; \text{S-GP-TS}) \leq 30\bar{a}\beta_T B \sqrt{\frac{2T\gamma_T}{\log(1 + \frac{1}{\tau})}} + (31\beta_T + \alpha_T)\epsilon TB + 15B\mathcal{B} + 2B \tag{23}$$

which can be simplified to

$$R(T; \text{S-GP-TS}) = \tilde{O}\left(\underline{a}\bar{a}(1 + c)B\sqrt{T\gamma_T} + \underline{a}^2(1 + c^2)\epsilon TB\right). \tag{24}$$

$\square$

## B.2 Proof of Lemma 1

It remains to prove the concentration inequality for the approximate statistics given in Lemma 1.

*Proof of Lemma 1.* By triangle inequality we have

$$
\begin{aligned}
|f(x) - \tilde{\mu}_t(x)| &\leq |f(x) - \mu_t(x)| + |\tilde{\mu}_t(x) - \mu_t(x)| \\
&\leq |f(x) - \mu_t(x)| + c_t\sigma_t(x) \quad \text{by Assumptions 4.}
\end{aligned}
$$

From Theorem 2 of [3], with probability at least $1 - \delta$,

$$f(x) - \mu_t(x) \leq \left(B + R\sqrt{2(\gamma_t + 1 + \log(1/\delta))}\right)\sigma_t(x).$$

Thus,

$$
\begin{aligned}
|f(x) - \tilde{\mu}_t(x)| &\leq \left(B + R\sqrt{2(\gamma_t + 1 + \log(1/\delta))}\right)\sigma_t(x) + c_t\sigma_t(x) \\
&\leq \underline{a}_t(B + R\sqrt{\frac{2\ln(1/\delta)}{\tau}} + c_t)(\tilde{\sigma}_t(x) + \epsilon_t),
\end{aligned}
$$

where the last inequality holds by Assumption 3. $\square$

## B.3 Proof of Proposition 1

Here, we use $\tilde{\mu}_t$ and $\tilde{\sigma}_t$ to specifically denote the approximate posterior mean and the approximate posterior standard deviations of the decomposed sampling rules (4) and (5) in contrast to Sec. 5.1 where we used the notation more generally for any approximate model. We also use $\mu_t^{(s)}$ and $\sigma_t^{(s)}$ to refer to the posterior mean and the posterior standard deviation of SVGP models, and $\mu^{(w)}$ and $\sigma^{(w)}$ to refer to the priors generated from an $M-$truncated feature vector. For the approximate posterior mean, we have $\tilde{\mu}_t = \mu_t^{(s)}$. However, the approximate posterior standard deviations $\sigma^{(s)}$ and $\tilde{\sigma}$ are not the same.

By the triangle inequality we have

$$|\tilde{\sigma}_t(x) - \sigma_t(x)| \leq |\tilde{\sigma}_t(x) - \sigma_t^{(s)}(x)| + |\sigma_t^{(s)}(x) - \sigma_t(x)|. \tag{25}$$

For the first term, following the exact same lines as in the proof of Proposition 7 in [24], we have

$$|\tilde{\sigma}_t^2(x) - \sigma_t^{(s)^2}(x)| \leq C_1 m_t |\sigma^2(x) - \sigma^{(w)^2}(x)| \tag{26}$$

where $C_1 = \max_{1 \leq t \leq T}(1 + ||K_{\mathbf{Z}_{m_t}, \mathbf{z}_{m_t}}^{-1}||_{C(\mathcal{X}^2)})$. [24] proceed to upper bound $|\sigma^2(x) - \sigma^{(w)^2}(x)|$ by a constant divided by $\sqrt{M}$. We use a tighter bound based on feature representation of the kernel. Specifically from definition of $\delta_M$ we have that

$$
\begin{aligned}
|\sigma^2(x) - \sigma^{(w)^2}(x)| &\leq \sum_{i=M+1}^{\infty} \lambda_i \bar{\phi}_i^2 \\
&= \delta_M,
\end{aligned}
\tag{27}
$$

which results in the following upper bound

$$|\tilde{\sigma}_t^2(x) - \sigma_t^{(s)^2}(x)| \leq C_1 m_t \delta_M. \tag{28}$$

For the standard deviations we have

$$
\begin{aligned}
|\tilde{\sigma}_t(x) - \sigma_t^{(s)}(x)| &= \sqrt{|\tilde{\sigma}_t(x) - \sigma_t^{(s)}(x)|^2} \\
&\leq \sqrt{|\tilde{\sigma}_t(x) - \sigma_t^{(s)}(x)||\tilde{\sigma}_t(x) + \sigma_t^{(s)}(x)|} \\
&= \sqrt{|\tilde{\sigma}_t^2(x) - \sigma_t^{(s)^2}(x)|^2} \\
&\leq \sqrt{C_1 m_t \delta_M},
\end{aligned}
\tag{29}
$$

where the first inequality holds because $|\tilde{\sigma}_t(x) - \sigma_t^{(s)}(x)| \leq |\tilde{\sigma}_t(x) + \sigma_t^{(s)}(x)|$ for positive $\tilde{\sigma}_t(x)$ and $\sigma_t^{(s)}(x)$.

We can efficiently bound the error in the SVGP approximation based on the convergence of SVGP methods. Let us first focus on the inducing features. It was shown that (Lemma 2 in [26]), for the SVGP with inducing features

$$\text{KL}\left(\text{GP}(\mu_t, \sigma_t), \text{GP}(\mu_t^{(s)}, k_t^{(s)})\right) \leq \frac{\theta_t}{\tau}. \tag{30}$$

where $\text{GP}(\mu_t, \sigma_t)$ and $\text{GP}(\mu_t^{(s)}, k_t^{(s)})$ are the true and the SVGP approximate posterior distributions at time $t$, and KL denotes the Kullback-Leibler divergence between them. On the right hand side, $\theta_t$ is the trace of the error in the covariance matrix. Specifically, $\theta_t = \text{Tr}(E_t)$ where $E_t = K_{\mathbf{X}_t, \mathbf{X}_t} - K_{\mathbf{Z}_t, \mathbf{X}_t}^{\mathsf{T}} K_{\mathbf{Z}_t, \mathbf{z}_t} K_{\mathbf{Z}_t, \mathbf{X}_t}$. Using the Mercer expansion of the kernel matrix, [26] showed that $[E_t]_{i,i} = \sum_{j=m_t+1}^{\infty} \lambda_j \phi_j^2(x_i)$. Thus

$$
\begin{aligned}
\theta_t &= \sum_{i=1}^{t} \sum_{j=m+1}^{\infty} \lambda_j \phi_j^2(x_i) \\
&\leq t \sum_{j=m_t+1}^{\infty} \lambda_j \bar{\phi}_j^2 \\
&= t \delta_{m_t}
\end{aligned}
\tag{31}
$$

Thus,

$$\text{KL}\left(\text{GP}(\mu_t, \sigma_t), \text{GP}(\mu_t^{(s)}, k_t^{(s)})\right) \leq \kappa_t/2. \tag{32}$$

where $\kappa_t = 2tB\delta_m/\tau$ that is determined by the number of current observations. In comparison, [26] proceed by introducing a prior distribution on $x_i$ and bounding $[E_t]_{i,i}$ differently.

For the case of inducing points drawn from an $\epsilon_0$ close k-DPP distribution, similarly following the exact lines as [26] except for the upper bound on $[E_t]_{i,i}$, with probability at least $1 - \delta$, (32) holds with $\kappa_t = \frac{2tB(m_t+1)\delta_{m_t}}{\delta\tau} + \frac{4tB\epsilon_0}{\delta\tau}$ where $\epsilon_0$ that is determined by the number of current observations.

In addition, if the KL divergence between two Gaussian distributions is bounded by $\kappa_t/2$, we have the following bound on the means and variances of the marginals [Proposition 1 in [26]]

$$
\begin{aligned}
|\mu_t^{(s)}(x) - \mu_t(x)| &\leq \sigma_t(x)\sqrt{\kappa_t}, \\
|1 - \frac{\sigma_t^{(s)2}(x)}{\sigma_t^2(x)}| &\leq \sqrt{3\kappa_t},
\end{aligned}
\tag{33}
$$

which by algebraic manipulation gives

$$
\sqrt{1 - \sqrt{3\kappa_t}}\sigma_t(x) \leq \sigma_t^{(s)}(x) \leq \sqrt{1 + \sqrt{3\kappa_t}}\sigma_t(x)
\tag{34}
$$

Combining the bounds on $\sigma_t^{(s)}$ with (29), we get

$$
\sqrt{1 - \sqrt{3\kappa_t}}\sigma_t(x) - \sqrt{C_1 m_t \delta_M} \leq \tilde{\sigma}_t(x) \leq \sqrt{1 + \sqrt{3\kappa_t}}\sigma_t(x) + \sqrt{C_1 m_t \delta_M}
$$

Comparing this bound with Assumption 3, we have $\underline{a}_t = \frac{1}{\sqrt{1 - \sqrt{3\kappa_t}}}$, $\bar{a}_t = \sqrt{1 + \sqrt{3\kappa_t}}$, and $\epsilon_t = \sqrt{C_1 m_t \delta_M}$. Also, since $\mu_t^{(s)} = \tilde{\mu}_t$, comparing (33) with Assumption 4, we have $c_t = \sqrt{\kappa_t}$. ☐

### B.4  Proof of Theorem 2

In Theorem 1, we proved that

$$
R(T, B; f) = O\left(\underline{a}\bar{a}BR\sqrt{d\gamma_T(\gamma_{TB} + \log(T))T\log(T)} + \underline{a}\epsilon TBR\sqrt{d(\gamma_{TB} + \log(T))\log(T)}\right)
$$

We thus need to show that $\underline{a}\bar{a}$ is a constant independent of $T$ and $\underline{a}\epsilon$ is small so that the second term is dominated by the first term.

In the case of Matérn kernel, $\lambda_j = O(j^{-\frac{2\nu+d}{d}})$ implies that $\delta_m = O(m^{-\frac{2\nu}{d}})$. Under sampling rule (4), we select $\delta = \frac{1}{T}$ and $\epsilon_0 = \frac{1}{T^2 \log(T)}$ in Proposition 1. We thus need $\kappa_T = O(T^2 m_T \delta_{m_T})$ and $\epsilon_T\sqrt{T} = O(\sqrt{m_T \delta_M T})$ be sufficiently small constants. That is achieved by selecting $m_T = T^{\frac{2d}{2\nu-d}}$ and $M = T^{\frac{(2\nu+d)d}{2(2\nu-d)\nu}}$.

Under sampling rule (5), we need $\kappa_T = O(T\delta_{m_T})$ and $\epsilon_T\sqrt{T} = O(\sqrt{m_T \delta_M T})$ be sufficiently small constants. That is achieved by selecting $m_T = T^{\frac{d}{2\nu}}$ and $M = \frac{(2\nu+d)d}{4\nu^2}$.

In the case of SE kernel, $\lambda_j = O(\exp(-j^{\frac{1}{d}}))$ implies that $\delta_m = O(\exp(-m^{\frac{1}{d}}))$. Under sampling rule (4), we select $\delta = \frac{1}{T}$ and $\epsilon_0 = \frac{1}{T^2 \log(T)}$ in Proposition 1. We thus need $\kappa_T = O(T^2 m_T \delta_{m_T})$ and $\epsilon_T\sqrt{T} = O(\sqrt{m_T \delta_M T})$ be sufficiently small constants. That is achieved by selecting $m_T = (\log(T))^d$ and $M = (\log(T))^d$. We obtain the same results under sampling rule (5) where we need $\kappa_T = O(T\delta_{m_T})$ and $\epsilon_T\sqrt{T} = O(\sqrt{m_T \delta_M T})$ be sufficiently small constants. ☐

## C   Additional Experiments and Experimental Details

In Section 6, we tested S-GP-TS across popular synthetic benchmarks from the BO literature. We considered the Shekel, Hartmann and Ackley (see Figure 3) functions, each contaminated by Gaussian noise with variance $0.1, 0.5$ and $0.5$, respectively. Note that for Hartmann and Ackley, we chose our observation noise to be an order of magnitude larger than usually considered for these problems in order to demonstrate the suitability of S-GP-TS for controlling large optimization budgets (as required to optimize these highly noisy functions). We now provide explicit forms for these synthetic functions and list additional experimental details left out from the main paper.

**Shekel function**. A four-dimensional function with ten local and one global minima defined on $\mathcal{X} \in [0, 10]^4$:

$$f(\mathbf{x}) = -\sum_{i=1}^{10} \left( \sum_{j=1}^{4} (x_j - A_{j,i})^2 + \beta_i \right)^{-1},$$

where

$$\beta = \begin{pmatrix} 1 \\ 2 \\ 2 \\ 4 \\ 4 \\ 6 \\ 3 \\ 7 \\ 5 \\ 5 \end{pmatrix} \quad \text{and} \quad A = \begin{pmatrix} 4 & 1 & 8 & 6 & 3 & 2 & 5 & 8 & 6 & 7 \\ 4 & 1 & 8 & 6 & 7 & 9 & 3 & 1 & 2 & 3.6 \\ 4 & 1 & 8 & 6 & 3 & 2 & 5 & 8 & 6 & 7 \\ 4 & 1 & 8 & 6 & 7 & 9 & 3 & 1 & 2 & 3.6 \end{pmatrix}.$$

**Ackley function**. A five-dimensional function with many local minima surrounding a single global minima defined on $\mathcal{X} \in [-2, 1]^5$:

$$f(\mathbf{x}) = -20 \exp \left( -0.2 * \sqrt{\frac{1}{4} \sum_{i=1}^{d} x_i^2} \right) - \exp \left( \frac{1}{4} \sum_{i=1}^{4} \cos(2\pi x_i) \right) + 20 + \exp(1).$$

**Hartmann 6 function**. A six-dimensional function with six local minima and a single global minima defined on $\mathcal{X} \in [0, 1]^6$:

$$f(\mathbf{x}) = -\sum_{i=1}^{4} \alpha_i \exp \left( -\sum_{j=1}^{6} A_{i,j} (x_j - P_{i,j})^2 \right),$$

where

$$A = \begin{pmatrix} 10 & 3 & 17 & 3.5 & 1.7 & 8 \\ 0.05 & 10 & 17 & 0.1 & 8 & 14 \\ 3 & 3.5 & 1.7 & 10 & 17 & 8 \\ 17 & 8 & 0.05 & 10 & 0.1 & 14 \end{pmatrix}, \qquad \alpha = \begin{pmatrix} 1 \\ 1.2 \\ 3 \\ 3.2 \end{pmatrix},$$

$$P = 10^{-4} \begin{pmatrix} 1312 & 1696 & 5569 & 124 & 8283 & 5886 \\ 2329 & 4135 & 8307 & 3736 & 1004 & 9991 \\ 2348 & 1451 & 3522 & 2883 & 3047 & 6650 \\ 4047 & 8828 & 8732 & 5743 & 1091 & 381 \end{pmatrix}.$$

For all our synthetic experiments (both for S-GP-TS and the baseline BO methods), we follow the implementation advice of [66] regarding constraining length-scales (to stabilize model fitting) and by maximizing acquisition functions (and Thompson samples) using L-BFGS [67] starting from the best location found across a random sample of $500 * d$ locations (where $d$ is the problem dimension). Our SVGP models are fit with an ADAM optimizer [65] with an initial learning rate of 0.1, ran for at most 10,000 iterations but with an early stopping criteria (if 100 successive steps lead to a loss less that 0.1). We also implemented a learning rate reduction factor of 0.5 with a patience of 10. Our implementation of the GIBBON acquisition function follows [10] and is built on 10 Gumbel samples built across a grid of 10,000 *$d$ query points. For BO's initialization step, our S-GP-TS models are given a single random sample of the same size as the considered batches and standard BO routines are given $d + 4$ initial samples (again following the advice of [66]). The function evaluations required for these initialization are included in our Figures.

## C.1 S-GP-TS on the Ackley Function

To supplement the synthetic examples included in the main body of the paper, we now consider the performance of S-GP-TS when used to optimize the challenging Ackley function, defined over 5 dimensions and under very high levels of observation noise (Gaussian with variance 0.5). The Ackley function (in 5D) has thousands of local minima and a single global optima in the centre. As this global optima has a very small volume, achieving high precision optimization on this benchmark requires high levels of exploration (akin to an active learning task). Figure 3 demonstrates the performance of S-GP-TS on the Ackley benchmark, where we see that S-GP-TS is once again able to find solutions with lower regret than the sequential benchmarks and effectively allocate batch resources. In contrast to our other experiments, where the K-means inducing point selection routine significantly outperforms greedy variance reduction, our Ackley experiment shows little difference between the different inducing point selection routines. In fact, greedy variance selection slightly outperforms selection by k-means. We hypothesize that the strong repulsion properties of DPPs (as approximated by greedy variance selection) are advantageous for optimization problems requiring high levels of exploration.

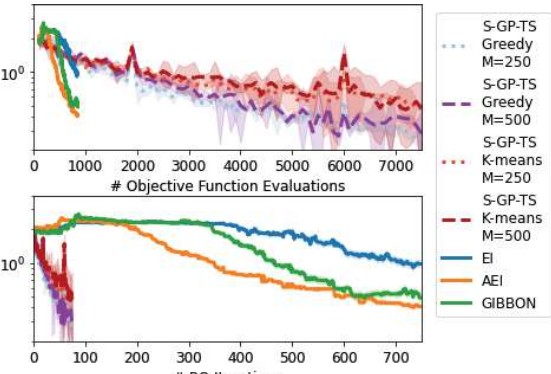

Figure 3: Simple regret on 5D Ackley function. The best S-GP-TS approaches are able to efficiently allocate additional optimization budgets to achieve lower final regret than the sequential baselines. When considering regret with respect to the BO iteration (bottom panels,idealised parallel setting), S-GP-TS achieves low regret in a fraction of the iterations required by standard BO routines. For this task, the choice of inducing point selection strategy (and number of inducing points) is not as crucial as for our other synthetic benchmarks, however, greedy variance selection provides a small improvement over selection by k-means.

## C.2 A Comparison of S-GP-TS with other batch BO routines

To accompany Figures 1 and 3 (our comparison of S-GP-TS with sequential BO routines), we also now compare S-GP-TS with popular batch BO routines. Once again, we stress that these existing BO routines do not scale to the large batch sizes that we consider for S-GP-TS, and so we plot their performance for $B = 25$ (a batch size considered large in the context of these exiting BO methods). We consider two well-known batch extensions of EI: Locally Penalized EI [LP, 8] and the multi-point EI (known as qEI) of [7]. We also consider with a recently proposed batch information-theoretic approach known as General-purpose Information-Based Bayesian OptimizatioN [GIBBON, 10]. The large optimization budgets considered in these problems prevent our use of batch extensions of other popular but high-cost acquisition functions such as those based on knowledge gradients [9] or entropy search [13]. Figure 4 compares our S-GP-TS methods (B=100) with the popular batch routines (B=25), where we see that S-GP-TS achieves lower regret than existing batch BO methods for our most noisy synthetic function (Hartmann).

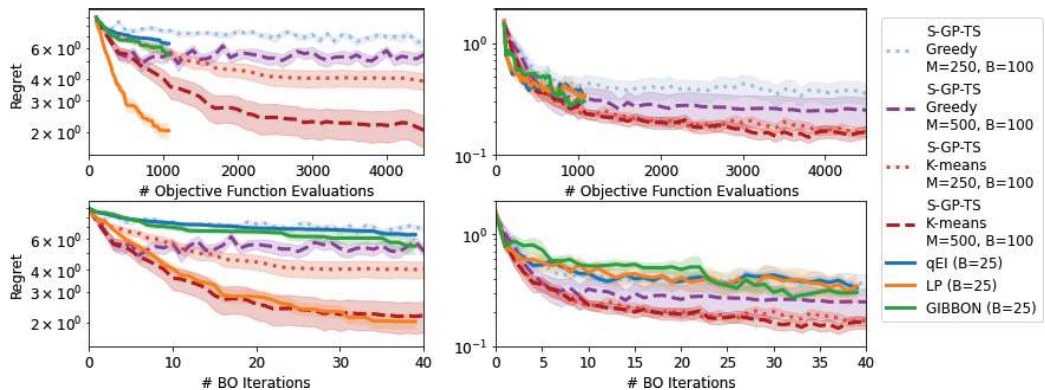

Figure 4: Simple regret on Shekel (4D, left) and Hartmann (6D, right) as a function of either the number of evaluations (top) or BO iterations (bottom). S-GP-TS methods are ran for batches of size $B = 100$ and the batch BO methods for batches of size $B = 25$. We see that S-GP-TS is particularly effective when performing the batch optimization of particularly noisy functions (Hartmann), exceeding the regret of the batch baselines. In our synthetic benchmark with low observation noise (Shekel), S-GP-TS is less efficient in terms of individual function evaluations, however, S-GP-TS 's ability to control larger batches means that it can match the performance of the highly perfomant LP with respect to the number of BO iterations (the idealised parallel setting).