# OpenReview forum: "Scalable Thompson Sampling using Sparse Gaussian Process Models"
_NeurIPS.cc/2021/Conference — NeurIPS 2021 Poster_

### Official Review · Reviewer_Psad · 2021-07-15

**Rating:** 6
**Confidence:** 2

**Summary:**

The work is dedicated to a regret vs computational complexity analysis of scalable Thompson sampling technique based on the ideas of Wilson et al, ICML 2020. Theoretical bounds of regret for the widely used Matern and squared exponential kernels are derived and the computational complexity is estimated. The conclusion is drawn that the scalable Thompson sampling technique achieves the same order of regret asymptotics as exact Thomspon sampling with substantially reduced complexity. Empirical tests on several benchmark functions and one real world problem are performed where the scalable Thompson sampling technique turns out to be on par with existing baselines.

**Limitations And Societal Impact:**

Some limitations are addressed. The societal impact is adequately addressed.

**Main Review:**

Theoretical results of the paper are interesting. They support the new scalable approximate Thompson sampling technique by confirming that in some particular regime (i.e. number of features, number of inducing variables) it attains the regret bound of the exact analog. The empirical study is rather scarce. Considering more benchmark functions would be beneficial as well as benchmarking runtimes and quantifying uncertainty for the real world problem. Although the delicacy of these is understandable because of implementation dependency of runtimes and the reported high computational demand of the experiment that hinders uncertainty quantification. Another point to mention is that theoretical analysis and the studied empirical setting do not agree: the former uses Mercer theorem expansions and minimizes acquisitions over a discrete set while the latter uses random Fourier features and minimizes acquisitions by a (repeated) gradient descent. This illustrates the fact that the theoretical setting studied differs from the setting that actually usually arises in applications and somewhat impairs the significance of this work.

**Warning**: due to time constraints and large number of papers to review I was unable to check the proofs and calculations of the paper. Since those are the main contributions I write this review assuming those are correct but retaining a very low level of confidence.

# Originality
The paper presents original theoretical bounds of regret for the new scalable approximate Thompson sampling technique and the basic supporting empirical study.

# Quality
See warning above on the theoretical side. The extended empirical study would be beneficial.

# Clarity
The paper is well-written.

# Significance
The results seem novel and useful for applications. As a byproduct of the theoretical study authors identify the regime of the scalable approximate Thompson sampling that is efficient and attains the regret asymptotic order of the exact analog. Probably the main problem is that the theoretical study does not align with the empirical one (uses a different setting).

# Remarks
- Lines 78–79. It is worth mentioning that the usual definition of sub-Gaussianity allows for nonzero mean and contains a multiplicative constant before the exponential function. Furthermore, it seems unnecessary and a bit confusing to introduce the sub-Gaussianity assumption for noise and then working with Gaussian noise. In my opinion, just assuming the Gaussianity of noise would be fairer and clearer.
- Line 94. This line refers the reader to the appendix for details on RKHS. After a quick look over the appendix I could not find a discussion on RKHS. If it does actually exist, a more explicit reference would help.
- Equation (2). $\tau$ is not defined/explained in text.
- Line 120. Probably “is given” -> “are given”.
- Line 130. I suggest doing something like “we can decompose our chosen kernel”->“we can usually decompose our chosen kernel”. After all, we can decompose only sufficiently regular kernels.
- Lines 170 and 178. It would be useful to elaborate on why the formulas are what they are, i.e. why do we have $\alpha_t (u_t - m_t) + m_t$ instead of just $\alpha_t u_t.$ Additionally, it would be useful to expand on why you consider two different sampling rules (given by (4) and (5)) and what are their benefits relative to each other.
- Line 194. “assumption”->”assumptions”.
- Equation after line 196. The notation for constant $\varepsilon_t$ overloads the notation for noise.
- Line 205. “$\tilde{u}_t (\tilde{\sigma}_t(x) + \varepsilon_t)$” -> “$\tilde{u}_t(\delta) (\tilde{\sigma}_t(x) + \varepsilon_t)$” because otherwise it looks like $\tilde{u}_t$ is evaluated at $\tilde{\sigma}_t(x) + \varepsilon_t$ rather than multiplied by it.
- Theorem 1. It would be useful to elaborate on the fact that the seemingly linear term in this regret will be subsequently removed by the very small value of $\varepsilon$.
- Line 222. Probably “Decomposed”->”Decoupled”.
- Line 350. Probably “$m_t$” -> “$m$”.


**Time Spent Reviewing:**

8

---

> ### Author Response · Authors · 2021-08-10
> **Response to Reviewer Psad**
>
> Thank you for the predominantly positive review, stressing the novelty and importance of our theoretical work and the clarity of our exposition. We now address your comments about our experimental study.
>
> *“The empirical study is rather scarce.”*
>
> As the primary contribution of our work is theoretical, we do not feel that additional experiments (over our four synthetic and real-world experiments) are needed to prove the utility of our method. Our empirical study is already more expansive than most BO theory papers, which typically consider only synthetic problems (e.g. the celebrated Srinivas et al. [2009]).
>
> *“Probably the main problem is that the theoretical study does not align with the empirical one (uses a different setting)”*
>
> As is usually the case (e.g. Srinivas et al. [2009]) our regret-based analysis applies to a version of the algorithm that is slightly different to a practically viable BO method.  Rather than focusing on recreating our algorithm exactly in a very limited setting (e.g. a 1-d RBF kernel for which we can calculate inducing features exactly) that is of little interest to the BO community, we choose to demonstrate the practical strength and unprecedented scalability of S-GP-TS by investigating an implementation that could be used by practitioners. For example, we wished to investigate the performance of S-GP-TS in a real-world high-throughput BO setting, of which molecular design loops form the primary examples. Although our chosen kernel is not completely aligned with our theoretical results (i.e. it permits only a random decomposition rather than an inducing feature decomposition), we decided that the importance of this particular problem within the BO literature was sufficient to support moving away from our theory. The resulting algorithm is still well-aligned with our work through its sparse GP surrogate model and decoupled Thompson sampling. We decided that it was important to include this result to show that, after only a minor implementation change, S-GP-TS can successfully control two orders of magnitude larger optimization budgets than existing GP-based BO methods and match the performance of an established BNN-based method.
>
>
> Finally, thanks for spotting the typos. They have helped make the paper clearer in its final version. In particular, we referenced information about RHKS more clearly and expanded our discussions around the linear regret term in Theorem 1 and the relative differences of our two different sampling rules.
>
> - *“It is worth mentioning that the usual definition of sub-Gaussianity allows for nonzero mean and contains a multiplicative constant before the exponential function. Furthermore, it seems unnecessary and a bit confusing to introduce the sub-Gaussianity assumption for noise and then working with Gaussian noise. In my opinion, just assuming the Gaussianity of noise would be fairer and clearer.”*
>
> Thanks, this assumption is mainly for the rigor of the theoretical results because we are using the confidence intervals of [3]. We will add the phrase “zero-mean” to subGaussian.
>
> - *“Lines 78–79. It is worth mentioning that the usual definition of sub-Gaussianity allows for nonzero mean and contains a multiplicative constant before the exponential function. Furthermore, it seems unnecessary and a bit confusing to introduce the sub-Gaussianity assumption for noise and then working with Gaussian noise. In my opinion, just assuming the Gaussianity of noise would be fairer and clearer.”*
>
> We later found the RKHS assumption too specialized and perhaps generic which could be found in other places. We will fix this error by adding a reference.
>
> - *“Equation (2). is not defined/explained in text.”*
>
> $\tau$ here is the assumed variance of the surrogate GP model. We have made this clearer in the final version.
>
> - *“Line 130. I suggest doing something like “we can decompose our chosen kernel”->“we can usually decompose our chosen kernel”. After all, we can decompose only sufficiently regular kernels.”*
>
> Thanks for pointing this out. We will clarify that we are considering “Mercer kernels” which are the kernels that satisfy the assumptions of the Mercer theorem.
>
> - * “Lines 170 and 178. It would be useful to elaborate on why the formulas are what they are”*
>
> This step is to ensure only the covariance is scaled using $\alpha_t$ and not the mean. We have made this clearer in the paper.
>
> - *“why you consider two different sampling rules (given by (4) and (5))”*
>
> These two methods seem to be standard in the literature, for example see [26]. Thus we included both as the analyses are similar. While  sampling rule (4) gives a slightly better computation complexity (Table 1) theoretically, it is empirically  more difficult to implement (4) since accessing eigenfeatures require the Mercer decomposition of the used kernel, which are available only for certain kernels on  manifolds [37, 34], but limited to low dimensions for others [38, 39].
>
> - *“Theorem 1. It would be useful to elaborate on the fact that the seemingly linear term in this regret will be subsequently removed by the very small value of $\epsilon$ “*
>
> Thanks. We have elaborated on this in the paper.

---

> > ### Comment · Reviewer_Psad · 2021-08-26
> > **The Reviewer's Response**
> >
> > Thank you for your response! My score remains unchanged.

---

### Official Review · Reviewer_sgAN · 2021-07-15

**Rating:** 7
**Confidence:** 2

**Summary:**

The paper provides regret bounds for Thompson sampling based on scalable Gaussian processes. Under certain conditions, scalable GP Thompson sampling enjoys the same regret bound as the exact GP Thompson sampling.


**Limitations And Societal Impact:**

The authors address the limitations of the scalable GP Thompson sampling, the main of those being the curse of dimensionality. The authors do not address potential societal impact. As the work is largely a theoretical analysis, the societal impact will depend on practitioners applying scalable Thomspon sampling to real-world problems.

**Main Review:**

The work presents regret bounds for Thompson sampling based on scalable Gaussian processes (i.e. with the inducing point approximation) that are sampled with decoupled sampling (i.e. incurring yet another approximation). There is a proof for the given regret bounds that seems solid. The work is original and cites related work adequately.

The paper is clearly written, the mathematical details being hidden in the appendix. I have not checked all the details.

The results are important because the main conclusion is that scalable GP-based Thompson sampling enjoys, under some conditions, the same regret bound as exact GP Thompson sampling.

**Time Spent Reviewing:**

6

---

> ### Author Response · Authors · 2021-08-10
> **Response to Reviewer sgAN**
>
> Thank you for your positive comments, arguing for acceptance due to the originality and significance of our theoretical results. We hope you can please share your interest in the paper with the other reviewers in the discussion period.

---

### Official Review · Reviewer_52bF · 2021-07-18

**Rating:** 6
**Confidence:** 1

**Summary:**

The manuscript provides a theoretical analysis of Thompson sampling with the sparse Gaussian processes (S-GP-TS) for Bayesian optimization (BO) as sparse GPs have been successfully used to improve the scalability of Thompson sampling with the exact GP (GP-TS). The manuscript shows that TS with any (sparse) approximation GP can achieve the same regret order as the (exact) GP-TS with certain conditions. It also includes numerical experiments which shows empirical performance (i.e., the regret bound over iterations) of S-GP-TS.

**Limitations And Societal Impact:**

Yes. The authors addressed the limitations in Discussion section.

**Main Review:**

The manuscript does not proposes new methodologies but I think it has certain contributions. The manuscript provides theoretical background and guarantees for sparse GPs for Thompson sampling.

I think that only one real-world dataset is not enough to make readers convinced about the effectiveness of the proposed method in practice. I also think that the authors could include the computational complexity of each method (BNN-TS or S-GP-TS) per each evaluation.

Lines 136 and 137: the subscript t is omitted (i.e., \phy_m(x) and \Lambda_m)

Line 306: “BO iterations” is used without its definition.

**Time Spent Reviewing:**

5

---

> ### Author Response · Authors · 2021-08-10
> **Response to Reviewer 52bF**
>
> Thank you for the positive review. We now discuss your two primary comments.
>
> *“The manuscript does not propose new methodologies”*
>
> Our proposed method is a result of using sparse Gaussian process models for Thompson sampling and we agree with the reviewer that this is indeed a natural algorithm and is already popular in  the literature (e.g. Wilson et al. 2020). However, despite its popularity, this algorithm has never previously been given a formal theoretical treatment (see Lines 58-69 in our paper) or tested in the high-data regimes we use to motivate our work, i.e. high-throughput BO with large optimization budgets and big batches. Crucially for the importance of our work, existing theoretical results cannot be applied in this setting. In particular, the error in the approximate GP model invalidates all existing regret bounds. Therefore, our main contribution of an analysis of the regret bounds in the presence of this error (Theorem 1) is innovative and provides rigorous theoretical results for an established algorithm.
>
> *“I think that only one real-world dataset is not enough to make readers convinced about the effectiveness of the proposed method in practice.”*
>
> Although we have just one real-world experiment, it tackles an important and well-studied problem in the BO literature. In particular, we are able to show that our S-GP-TS can successfully control two orders of magnitude larger optimization budgets than existing GP-based BO methods and match the performance of an established BNN-based method. As the primary contribution of our work is theoretical, we do not feel that additional experiments (over our four synthetic and real-world experiments) are needed to prove the utility of our method.
>
> Finally, In response to your minor comments, we have added in the missing subscripts of Lines 136 and 137 and  defined “BO iterations” as the number of BO steps (i.e. the period over which we fit a surrogate model and allocate the next batch of points)

---

### Official Review · Reviewer_eTmV · 2021-07-25

**Rating:** 4
**Confidence:** 5

**Summary:**

The paper present a new approximate Bandit/Bayesian optimization algorithm named S-GP-TS based on Thompson sampling with Gaussian process (GP-TS). GP-TS is known to achieve good regret, it is also highly computationally inefficient. In particular it has two bottlenecks:
- computing the posterior on T historical points requires at least $O(T^2)$ and is not scalable
- even given the posterior, drawing a sample and evaluating it on N test points requires $O(N^2)$. This is true even if $T=0$, i.e. even evaluating an exact GP draw from prior on N points is computationally hard.

The paper leverages some old and some new results on speeding up these two bottlenecks to speed up the overall S-GP-TS algorithm:
- it replaces slow exact inference of the posterior with a sparse variational approximation, in two variants based on random features or inducing points
- it replaces slow exact sampling from the posterior with a recently proposed technique called decomposed sampling

Both of these are known techniques, but the authors take care in correctly matching the right variational approximation form and parameters with the right decomposed sampling form and parameters. This is not difficult but it is important and well explained in the paper. They also add batching in the most straightforward way for TS (draw a batch of B samples).

Once they have derived their algorithm, they proceed to provide sufficient conditions (mostly on the number and distribution of the random feature/inducing points) to guarantee that the algorithm achieve low regret. This analysis is essentially a direct combination of three existing analysis:
- "[3] Chowdhury and Gopalan" for the overall regret bounding framework, which adds the necessity of introducing a discretization of the action space
- "[26] Burt et al." for the accuracy bound of the sparse variational posterior. Normally this results could not be used as they do not hold for adaptive sequences, but since the authors are already operating on a fixed discretization following [3], they smartly take advantage of this fact to be able to apply the results of [24]
- "[24] Wilson et al." for the decomposed sampling, here once again thanks to the discretization the application is straightforward

The only extra technical issue that the authors resolve is handling additive error in their posterior estimate that was not considered often in the literature. However thanks to the sufficiently fast converging rate of the approximation this additive error does not require new tools, only a careful bookkeeping.

Overall the algorithm achieves the same rate as GP-TS. The only downside is the dependency on the batch size which is essentially linear (i.e. it would be almost equivalent to B instances operating in parallel and not sharing information).

The runtime is greatly reduced although the final complexity table (Table 1) could be accompanied by more explainations (more in the main review).

Finally the authors evaluate the methods empirically on large scale artificial and real world datasets. However the tuning of all the S-GP-TS is completely decoupled from theory, making it a less convincing evaluation. The authors also report simple regret instead of cumulative regret (the focus of the paper) and only report iterations and not wall-clock runtime.

**Limitations And Societal Impact:**

The authors discuss a bit the main limitations of the algorithms in the end, but these limitations should have been considered more throughout the paper (the necessity of the discretization mostly). The open question on how to choose inducing point is also not very in-depth (it misses developments in $k$-DPP sampling from 2019).

BO can have significant societal impact, but as this is mostly a theory paper it is ok to not discuss them.

**Main Review:**

Overall I like the idea of the paper as combining two solid and popular techniques (GP-TS and SVGP) with the newly introduced decoupled sampling. However the theoretical and algorithmic part seems almost completely incremental, with no clearly highlighted contribution over the secondary one of handling of additive error.

This could have been compensated with a good experimental evaluation, but although the authors positively focused on the latest algorithm and real-world, large scale dataset they also diverge significantly from the main premise of the paper: reducing wall-clock time (at least for the GP part) while guaranteeing small cumulative regret. The experiments do not report runtime nor cumulative regret, and are run with parameters that do not guarantee low regret.

Overall this looks like a highly polished, but incremental paper. The theoretical impact is medium, but since it is disconnected by the impact evaluated by the experiments it falls under the threshold.

More concrete ways in which the theoretical part of the paper could be improved:
1.  if I understood correctly, the discretization is not only necessary to guarantee correct maximization of the TS draw. Even in a domain where optimization is easy (e.g. linear kernel or 1d) the discretization needs to be there to make the selected points non-adaptive. Since the discretization scales as $t^{2d}$, the linear $N_t$ cost by itself would effectively negate any improvement due to the sparse approximation. It would be maybe better to explicitely target a finite action space (e.g. the library of molecules used in 6.2 is finite and fixed). Would this be enough to avoid having to concretely construct the discretization to run the algorithm?
2. Thm. 1 establishes a required value for $\alpha_t$, Proposition 1 is analyzed under $\alpha_t = 1$. This is not rigorous and should be corrected. It is also not clear how the authors plan to compute a sufficiently accurate value of $\gamma_t$ to be used in $\alpha$ with theoretical guarantees.
3. The whole paper is extremely focused on GP-TS, and almost completely ignores other important baselines in BO. Theoretically, at least UCB based algorithms, which actually achieve a lower regret and potentially runtime than S-GP-TS, should be mentioned and compared in terms of regret and runtime. Currently, they are only mentioned once in the conclusions. The experiments at least have EI, AEI and GIBBON, but again UCB is at least as popular as TS and should be included. Scalability is not an issue since as the authors comment, "[30] Calandriello et al." seems as scalable as S-GP-TS, and there seems to be a [follow-up version capable of batching available](https://github.com/luigicarratino/batch-bkb).
4. Table 1 is quite unreadable, and again it is not compared to any other scalable GP optimization method. Even not considering the lack of baselines, in most of the paper the authors use maximum information gain to express complexities, while here everything is expanded (e.g. the first $m_t$ seems to be $\gamma_t^2$ but it is not unclear. It would be good to 4.A) uniform somehow the various complexity so that they can be compared (e.g. it is hard to compare conditions on M for inducing point and features for the matern kernel), 4.B) add remarks explaining rates and pros/cons of each alternative, and 4.C) add a comparison with existing BO baselines.

The experiments also could be greatly improved, both to more closely match the theory and in general to be made a more rigorous evaluation:
1) $\alpha_t$ in the experiment should be set according to Thm. 1 and not set to 1. $m_t$ and $M$ should also try to satisfy the values dictated by theory. If not, they should be cross validated and all curves (also the sub-optimal ones) reported. Cross validation is not usually possible in real-world BO, but toy experiments are an ideal place for this kind of explorations.
2) Cumulative regret is the goal of the paper and it is not reported.
3) Wall clock or some form of similar evaluation should be reported. In general the golden rule would be to report your own computational resources and wall-clock (as recommended by the checklist), and take the baselines numbers from the original papers. In case the original papers use different dataset or do not report numbers, it is worth to reimplement and run at least one for comparison. If all else fails, one can substitute per-step theoretical complexities when wall-clock is not available.
4) Why do you compare BO iterations and function evaluations without including at least another batched baseline? Usually these quantities are compared to show that a batched algorithm does not have much more regret than a sequential one. If instead the goal is to compare convergence speed then the baselines should also be batched (as they are in the appendix). The procedure used to choose the batch size has also not been reported (cross-validated?).
5) In the main paper experiments S-GP-TS has access to almost 5k evaluation, and baselines to 750. Without wall-clock constraint this seems completely arbitrary, and with enough evaluations the baselines might outperform S-GP-TS.

Minor comments:
- Thm. 1 is more precise, and a $O(\sqrt{d})$ factor should also be present in Thm. 2 ($\gamma_T$ which is not a constant depends on it)
- recent $k$-DPP sampling algorithms (e.g. [the intermediate samplers from DPPy](https://dppy.readthedocs.io/en/latest/finite_dpps/exact_sampling.html#id26)) are not slow, and they are guaranteed to be roughly as efficient as S-GP-TS, at least for moderate $k$.

**Time Spent Reviewing:**

9

---

> ### Author Response · Authors · 2021-08-10
> **Response to Reviewer eTmV**
>
> Thank you for your thorough review of the paper. We appreciate the amount of time that has gone into this review. We will respond to your comments point by point.
>
> *“this additive error does not require new tools, only a careful bookkeeping.”*
>
> We believe our contribution in providing an analysis for a scalable TS using sparse GP models and decoupled sampling is more than putting 3 papers together. We have mentioned our contributions in the last paragraph of section 1 (lines 58-69). The complexities of an additive error in uncertainty (which may yield under exploration) cannot be handled by existing work and the analytical details can be found on page 16 of the supplementary material. Therefore, our main contribution of an analysis of the regret bounds in the presence of this error (Theorem 1) is innovative. Moreover, we also provide additional theoretical results that characterize the algorithmic parameters needed to guarantee the same order of regret bounds as in vanilla GP-TS. All the other reviewers stress the novelty of our convergence analysis.
>
> *“The only downside is the dependency on the batch size which is essentially linear”*
>
> Please notice that for a fair comparison in terms of both the batch size and number of samples we should consider $T’ =BT$ as the number of samples. In that case our regret bounds $\tilde{O}\sqrt{BT’\gamma_{T’}\gamma_{T’/B}}$ scale with a $\sqrt{B}$ multiplicative factor. That is $\sqrt{B}$ tighter than the trivial scaling with $B$. We will remark on this point after the regret bound.
>
>
> We now respond directly to the reviewer’s numbered theoretical points:
>
> 1. Yes, indeed a finite action space solves this issue in the sense that the next observation point is selected among these finite sets. We would like to emphasize that the optimization of the acquisition function (posterior GP sample in TS or UCB index in GP-UCB) is a multimodal optimization problem. All of the acquisition based algorithms (GP-UCB, GP-TS, GP-PI, GP-EI) need to optimize an acquisition function. The standard analytical approach is to neglect the cost of this optimization (the same for all the existing work to the best of our knowledge) and assume that the acquisition function can be efficiently optimized. The rationale behind this approach to GP bandits is that the optimization of $f$ is based on expensive observations from environment (the performance of a design, the error rate of a neural network model for certain hyper parameter configuration), while optimization of the acquisition function is a purely computational matter. Instead of assuming a perfect optimization of the acquisition function, we are giving more detailed and less-restrictive analytical guarantees through a discretization argument. This is also exactly with respect to the size of this discretization that we improve the $O(N^3)$ computational cost of vanilla GP-TS (that is the cost of sampling due to a Cholsky decomposition step) to $O(N)$, as a result of our decoupled sampling method.
>
> 2. Please notice the distinction between the posterior kernel of the GP model and the kernel from which we draw the Thompson sample. The covariance is scaled with $\alpha_t$ in the latter to ensure sufficient exploration as discussed in lines 170-175. This is standard and is required as in the analysis of vanilla GP-TS too [3]. Proposition 1 is concerned with the relation between the approximate and exact posterior variances. So we set $\alpha_t=1$. The approximation error is scaled with the same factor $\alpha_t$ which is taken into account  in the analysis of Theorem 1, and there is no discrepancy. Assuming a bound on $\gamma_t$ as well as the sub-Gaussianity parameter of noise and the RKHS norm of $f$, all are standard in the literature and are similar to vanilla GP-TS  and GP-UCB. This is not a specific shortcoming of our analysis.
>
> 3. Following your review, we expanded our current  discussion of Calandrello et al. [2020] in both a theoretical and practical context. We stress that the alternative method of Calandriello et al. [2020], although enjoying an improvement in scalability over standard BO with exact GP surrogate models, would still be prohibitively expensive for the large molecular search problem upon which we showed S-GP-TS to excel. In fact S-GP-TS scales better than this UCB-based approach for two key reasons. Firstly, Calandriello et al. [2020] require the maximization of standard acquisition functions, each query of which incurring an $O(Nm^2)$ cost (for the prediction of a sparse GP with $N$ datapoints and $m$ inducing points), whereas querying a decoupled sample from the SVGP (as required for S-GP-TS) is only an $O(Nm)$ cost (see Lines 157 and 164). The reason for this improvement in the case of S-GP-TS is that in addition to sparse update of the posterior model, we use the recently proposed method of decoupled sampling that results in an additional $m$ fold gain in the computational complexity. Secondly, where TS-based methods can cheaply allocate very large batches through its ability to be trivially parallelized (see Hernandez-Lobato et al. [2017] and our discussion from Lines 21-38), whereas Calandriello et al. [2020] require the sequential building of batches. We in fact did run this method for our smaller synthetic experiments when writing the paper, however its poor practical performance (as is often common for UCB-based approaches to BO, e.g. see Chapelle et al. [2011] or Kandasamy et al. [2018a]) meant that we chose not to include it in the submitted version. We have added these results back into the final version.
> 4. When specialized for the Matérn family of kernels, the number of inducing points in our analysis is in the same order as the m based on information gain in Calandriello et al. [2020]. As mentioned above we have extended our comparison with this paper.
> We would like to add a remark that Calandrello et al. [2020] borrow an adaptive matrix sketching method based on ridge leverage score for their sparse approximation of the posterior model. In contrast, we build our results based on the classic variational models dating back to Titsias et al. [2009]  which is the standard technique in the GP community. The comparison between these two methodologies is beyond the scope of our paper. An excellent recent paper has studied the connection and differences between these Bayesian and frequentist approaches to sparse approximations of kernel matrix inversion (Wild et al. [2021], "Connections and Equivalences between the Nyström Method and Sparse Variational Gaussian Processes").
>
>
> We respond to the reviewer’s general comment about the closeness of our experiments to our theoretical results. As is usually the case (e.g. Srinivas et al. [2009]) our regret-based analysis applies to a version of the algorithm that is slightly different to a practically viable BO method.  Rather than focusing on recreating our algorithm exactly in a very limited setting (e.g. a 1-d RBF kernel for which we can calculate inducing features exactly) that is of little interest to the BO community, we choose to demonstrate the practical strength and unprecedented scalability of S-GP-TS by investigating an implementation that could be used by practitioners. We now address your individual concerns:
>
> 1. To make a simple practical version of our algorithm, it is important that we do not have dynamic tunable parameters (like those that plague UCB-based approaches), so we make the simple recommendation of setting $\alpha=1$. Also, choosing a fixed number of inducing points (rather than following our theoretical conditions) is incredibly important for both controlling the computational overhead of the algorithm and for the efficiency of our computational implementation (i.e. avoiding TensorFlow retracing issues). We have provided clearer justification for these choices in the final version.
> 2. We are plotting average cumulative regret rather than simple regret. While average cumulative regret can also be referred to as simple regret, the simple averaging is often not the main distinguishing feature between the two. When simple regret is distinguished from cumulative regret, it is often to remove the exploitation part from the exploration-exploitation tradeoff in online learning problems. In simple regret, there is only an exploration phase where the best point is chosen at the end. In cumulative regret however the trajectory of the observations is also considered and the performance measure is a cumulative loss which gives rise to the exploration-exploitation tradeoff. In this sense, we are reporting average cumulative regret (it is averaged simply for better visibility) rather than simple regret. We will replace the term simple regret in the experiments with average cumulative regret to avoid the confusion. Thanks for pointing this out.
> 3. The entire selling point of our method is its scalability, and that it can be ran on large scale problems that other methods (e.g. standard BO baselines cannot). This is why we were unable to do like-for-like comparisons with the baselines. It does not make sense to compare wall-clock timings for sequential (or small batch) EI with our large batch S-GP-TS. These comparisons were just to provide context for the performance of S-GP-TS and demonstrate the potential of using the large parallel resources that previously could not be handled by BO.
> 4. We mainly answer this point above. Note that it is very uncommon to choose the ideal batch size for a BO loop as this is typically dictated by your experimental set-up (e.g. number of clusters or gene sequencing resources).
> 5. We reiterate that the baselines cannot handle as many evaluations as S-GP-TS (we found that our pipeline crashed once we surpassed 750 datapoints). Our experiments show that it is sometimes beneficial (i.e. for very noisy problems or complex search spaces) to perform many more than 750 evaluations, hence the importance of methods like S-GP-TS.

---

### Author Response · Authors · 2021-08-10
**Summary of reviews**

We thank all reviewers for conscientious and insightful readings of our paper, with unanimous assessment of our work as making theoretical contributions to the Bayesian Optimisation literature, providing a theoretically-justified scalable algorithm backed up with  empirical evidence and clear presentation.

There were some minor queries, mainly about the relationship between our theoretical results and experimental evaluation, which we address below.

---

### Decision · Program_Chairs · 2021-09-28

**Decision:**

Accept (Poster)

**Comment:**

I am ultimately recommending acceptance for this paper on the basis of the reviewer consensus that the theoretical contributions made in this paper are strong. Namely, this is the first paper I am aware of to (1) present a regret bound for Bayesian optimization with sparse GP models using any acquisition function at all, and (2) moreover, the theoretical results are specifically for a recent scalable approximation of Thompson sampling using sparse GPs, noting that Thompson sampling is one of those rare GP operations not made more computationally tractable by simply using sparse GPs alone.

However, I want to echo Reviewer eTmV's concerns about the choice of \alpha_{t}=1 in the experiments. Keep in mind, the actual acquisition function that you've demonstrated convergence for is *not* Thompson sampling as we would all understand it, or even an approximation thereof: it's a modification to Thompson sampling in which the (co-)variance envelope samples are drawn from shrinks over time. This is fine, and indeed there is obvious precedent for such a modification for theoretical purposes, because as you point out people who use UCB don't actually typically anneal the trade-off parameter. That said, Srinivas et al. absolutely did evaluate their mechanism with the annealed trade-off parameter: they did not evaluate a constant \beta_{t} but rather scaled it down by a factor of five, leaving a schedule for \beta_{t} that still annealed over time. I don't agree that setting \alpha_{t}=1 is at all analogous to their experimental evaluation.

While I completely understand that in practice, people will simply just use Thompson sampling (which has no parameter \alpha_{t}), there's frankly just no good justification for not at least presenting some results using an \alpha_{t} annealing schedule within a constant factor of the one suggested by your theory. You should include these results, if not in the main body of the paper then in the supplementary materials.

**Consistency Experiment:**

NeurIPS has a long history of experimentation. In 2014, NeurIPS ran an experiment in which 10% of submissions were reviewed by two independent committees to quantify the randomness in the review process. This year, we repeated a variant of this experiment to see how the quality of the review process has changed over time.  This paper was part of the experiment and was therefore assigned to two committees (consisting of reviewers, an Area Chair, and a Senior Area Chair) that reached independent decisions.  If both committees made the same recommendation, this recommendation was followed. If a single committee recommended acceptance, the paper was accepted (with the exception of a few cases in which the other committee identified what we considered a fatal flaw, e.g., an error in a key result).

Both committees reached the same decision: **Accept (Poster)**

The other committee assigned to the paper recommended **Accept (Poster)**.  You can find the other set of reviews, along with any follow up discussion with the authors here:
https://openreview.net/forum?id=mAiUwoBipv7